# Learning Associative Inference Using Fast Weight Memory

**Imanol Schlag**
The Swiss AI Lab IDSIA / USI / SUPSI
`imanol@idsia.ch`

**Tsendsuren Munkhdalai**
Microsoft Research
`tsendsuren.munkhdalai@microsoft.com`

**Jürgen Schmidhuber**
The Swiss AI Lab IDSIA / USI / SUPSI
`juergen@idsia.ch`

## Abstract

Humans can quickly associate stimuli to solve problems in novel contexts. Our novel neural network model learns state representations of facts that can be composed to perform such associative inference. To this end, we augment the LSTM model with an associative memory, dubbed *Fast Weight Memory* (FWM). Through differentiable operations at every step of a given input sequence, the LSTM *updates and maintains* compositional associations stored in the rapidly changing FWM weights. Our model is trained end-to-end by gradient descent and yields excellent performance on compositional language reasoning problems, meta-reinforcement-learning for POMDPs, and small-scale word-level language modelling.[1]

## 1 Introduction

Humans continually adapt in order to understand new situations in changing environments. One important adaptive ability is *associative inference* for composing features extracted from distinct experiences and relating them to each other (Schlichting & Preston, 2015; Gershman et al., 2015). Suppose Alice has shared with you pictures of her toddler. Later, at the office party, you see a man carrying the depicted toddler. Since the toddler yields a shared feature in two different contexts, it may be plausible to infer that the man is Alice's partner, without ever seeing him and Alice together. The ability to rapidly associate and bind together novel stimuli can help to derive knowledge systematically, in addition to the knowledge gained directly from observation.

Virtually all modern cognitive architectures applied to challenging artificial intelligence problems are based on deep artificial neural networks (NNs). Despite their empirical successes and theoretical generality, NNs tend to struggle to generalise in situations similar to the given example (Lake et al., 2017; Phillips, 1995; Lake & Baroni, 2017). This weakness becomes even more severe if the training and test data exhibit systematic differences (Atzmon et al., 2016; Agrawal et al., 2017). For example, during training, the man's representation might never be associated with the toddler's, but during testing, this association might be necessary to make a useful prediction. In problems where humans excel, this sort of inference is likely ubiquitous since data is often combinatorially complex in a way that observations used during training will likely cover just a small fraction of all possible compositions. Such a lack of productivity and systematicity is a long-standing argument against the use of NNs as a substrate of an artificial cognitive architecture (Fodor & Pylyshyn, 1988; Hadley, 1994; McLaughlin, 2009).

The hidden state of a neural model is a learned representation of the task-relevant information extracted from the input. To generalise to never-seen-before compositions of stimuli, the function which produces the state representation must be able to systematically construct *all possible* states. This requires a general and preferably differentiable method, such as the Tensor Product Representation (TPR; Smolensky (1990)). TPRs provide a general and differentiable method for embed-

---

[1]Source code and data used in this paper is available at github.com/ischlag/Fast-Weight-Memory-public

ding symbolic structures in vector spaces. A TPR state representation is constructed via the tensor product (i.e. the generalised outer-product) of learned component representations. Under certain constraints, such a mechanism guarantees a unique representation for every possible combination of components (Smolensky, 1990; 2012).

In this work, we augment a recurrent NN (RNN) with an additional TPR-like memory representation. To facilitate the learning of multi-step associative inference, the TPR memory can be queried multiple times in a row, allowing the model to chain together various independent associations. In contrast to previous work on fast weights, we apply our memory-augmented RNN to much longer sequences. This requires the model to *update* its associative memory. Furthermore, we demonstrate the generality of our method by applying it to meta-reinforcement learning and small scale language modelling problems.

In the next section, we cover related memory-augmented NNs. Section 3 describes the FWM in detail. Section 4 demonstrates the generality of our method through experiments in the supervised, self-supervised, and meta-reinforcement learning setting. The supervised-learning experiments in subsection 4.1 consist of a more challenging version of the bAbI dataset dubbed concatenated-bAbI or catbAbI. The meta-reinforcement learning experiment in section 4.2 demonstrates the FWM's ability to learn to explore a partially observable environment through its ability to perform associative inference. Finally, the self-supervised experiments in subsection 4.3 demonstrate that the FWM can compete with the state-of-the-art word-level language models on small benchmark datasets.

## 2 RELATED WORK

RNNs such as the Long Short-Term Memory (LSTM; Hochreiter & Schmidhuber (1997); Gers et al. (2000)) are in theory capable of implementing any algorithm (Siegelmann & Sontag, 1991). However, the linear growth of the hidden state of a fully connected RNN leads to quadratic growth in the number of trainable weights. Early work addressed this issue through the use of additional memory (Das et al., 1992; Mozer & Das, 1993) and differentiable fast weights (Schmidhuber, 1992; 1993). Recently, memory-augmented NNs have solved algorithmic toy problems (Graves et al., 2014; 2016) as well as reasoning and inference problems in synthetic and natural language (Weston et al., 2015b; Xiong et al., 2016).

Inspired by the random-access memory of computer architectures, a common approach is to incorporate a soft and differentiable lookup table into the NN model. Such slot-based memory matrices have shown to be difficult to train (Munkhdalai & Yu, 2017b) and require sophisticated mechanisms for the allocation and deallocation of memory (Csordas & Schmidhuber, 2019). The Transformer-XL (TXL; Dai et al. (2019)), an autoregressive language model variant of the Transformer (Vaswani et al., 2017), can be understood as a slot-based memory-augmented RNN where every new state is pushed into an immutable queue of finite size. Although it is recurrent, the layers of a transformer architecture are strictly forced to use inputs from a lower layer which limits its generality. Nevertheless, a sufficiently deep and well regularised TXL model has achieved state-of-the-art performance in large scale language modelling tasks.

A biologically more plausible alternative of increasing the memory capacity of NNs are fast-changing weights, i.e. stateful weights that can adapt as a function of its input. Non-differentiable fast weights or "dynamic links" have been published since 1981 (von der Malsburg, 1981; Feldman, 1982; Hinton & Plaut, 1987). Subsequent work showed that a regular network can be trained by gradient descent to control the fast weights of a separate network (Schmidhuber, 1992) or of itself (Schmidhuber, 1993) in an end-to-end differentiable fashion. Recently, fast weights have made a comeback and achieved good results in small toy problems where regular NNs fall short (Ba et al., 2016a; Schlag & Schmidhuber, 2017; Munkhdalai & Yu, 2017a; Pritzel et al., 2017; Ha et al., 2017; Zhang & Zhou, 2017; Miconi et al., 2018; 2019; Schlag & Schmidhuber, 2018; Munkhdalai et al., 2019; Bartunov et al., 2020).

Most memory-augmented NNs are based on content-based or key-based lookup mechanisms. An alternative to the storage of patterns in a lookup table is the idea that patterns are reconstructed through the implicit iterative minimisation of an energy function, such as in the classical Hopfield network (Steinbuch, 1961; Willshaw et al., 1969; Hopfield, 1982; Kanerva, 1988) or the modern Hopfield network (Krotov & Hopfield, 2016; Demircigil et al., 2017; Ramsauer et al., 2020). This is

often described as an *auto-associative* type of memory as it reconstructs a previously stored pattern that mostly resembles the current pattern. A much less studied variation is the *hetero-associative* memory (see e.g. Kosko (1988)) where the retrieved pattern is different from the input pattern. This is more relevant for our use case. We aim to train an LSTM to **construct**, **maintain**, and **edit** its associative memory. The ability to edit Hopfield networks partially is not very well studied. For this reason, we employ a simple (multi-)linear hetero-associative memory as it is more closely related to the theory of TPRs (whose manipulation is well understood) and because the association is retrieved in a single step.

Our work directly builds on two examples of differentiable fast weight memories: the TPR-RNN by Schlag & Schmidhuber (2018) and the Metalearned Neural Memory (MNM) by Munkhdalai et al. (2019). The TPR-RNN is a sentence-level model for reasoning on text. It achieves excellent results on the regular bAbI tasks but it underperforms on word-level bAbI (Schlag et al., 2019) or algorithmic toy problems (Le et al., 2020). In contrast, the MNM is a word-level model which augments the LSTM with a fully-connected multi-layer feed-forward network as its memory and trains it using a meta-learning objective. Both, MNM and TPR-RNN were developed on the regular bAbI dataset which only contains short sequences and does not require the model to remove deprecated associations from its memory. In this work, we train on an infinite sequence of bAbI stories where our FWM achieves excellent performance and improves over MNM. We further demonstrate strong performance in small-scale language modelling and meta reinforcement-learning which demonstrates the generality of our contribution.

## 3 PROPOSED METHOD

Our FWM is a fast-changing, multi-linear map which is controlled by a slowly-changing, non-linear LSTM. The slow weights of the LSTM are regular NN weights which are updated during training by gradient descent. In contrast, the fast weights of the FWM are updated by the LSTM at every step of the input sequence through a Hebb-like differentiable mechanism. This allows the FWM function to change rapidly even during testing—hence the name *fast* weights. Along with updating the fast weights, the LSTM also generates a memory query which is used to retrieve information that was previously stored. The retrieved information then becomes part of the model's output.

### 3.1 THE FAST WEIGHT MEMORY

Given a sequence of tokens $\mathbf{x} = (x_1, ..., x_T)$ from a vocabulary $\mathbb{V}$, the task of language modelling is to train a model which maximizes the joint probability $p(\mathbf{x})$ which we factorize autoregressively $p(x_{1:T}) = \prod_{t=1}^{T} p(x_t | x_{0:t-1})$ where $x_0$ is an artificial start token.[2] In this work, we train an RNN model to encode the input sequence $\mathbf{x}_{1:t}$ into $\boldsymbol{h}_t$, the hidden state of the LSTM, and $\boldsymbol{F}_t$, the fast weight tensor of the FWM, to maximize the probability of the next token $x_{t+1}$.

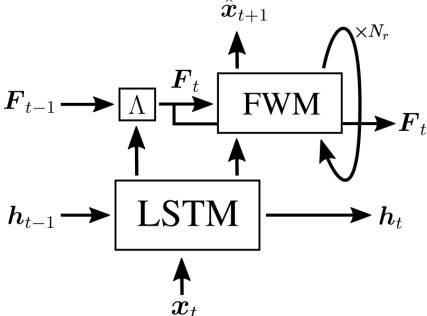

Figure 1: A simplified illustration of our proposed method where $\Lambda$ refers to the write mechanism described in section 3.1.1. $\boldsymbol{F}_t$ are the recurrent weights of the FWM which have been generated by the LSTM. The LSTM is a regular slow RNN. The residual connection between the FWM and the LSTM is not depicted.

At step $t$ of the input sequence, the input token $x_t$ is embedded in a $d_E$-dimensional vector space using a lookup table $\boldsymbol{e}_t = \text{embedding}(x_t)$. An LSTM with $d_{\text{LSTM}}$ hidden units encodes the sequence of embedded tokens into a fixed size vector representation $\boldsymbol{h}_t = \text{LSTM}(\boldsymbol{e}_t, \boldsymbol{h}_{t-1})$. The probability distribution over the next token $\hat{\boldsymbol{x}}_{t+1} = \text{softmax}(\boldsymbol{W}^{(s)}(\boldsymbol{h}_t + \text{FWM}(\boldsymbol{h}_t, \boldsymbol{F}_t))$ where $\boldsymbol{F}_t \in \mathbb{R}^{d_{\text{FWM}} \times d_{\text{FWM}}^2}$ are the fast weights of the FWM at step $t$ and $\boldsymbol{W}^{(s)} \in \mathbb{R}^{|\mathbb{V}| \times d_{\text{LSTM}}}$. Note that the fast weight matrix $\boldsymbol{F}_t$ is a reshaped third-order tensor $\mathbf{F}_t \in \mathbb{R}^{d_{\text{FWM}} \times d_{\text{FWM}} \times d_{\text{FWM}}}$. This allows us to describe third-order tensor operations using matrix multiplications. We'll now describe in detail the FWM function and how its fast weights are updated.

---

[2]We use the notation $\mathbf{x}_{1:t}$ to refer to the sequence $(x_1, x_2, ..., x_t)$.

### 3.1.1 WRITING

The FWM is updated at every step $t$ using the write mechanism described in this section. To this end, we extract from the hidden state $\boldsymbol{h}_t$: the write strength $\beta$ (a scalar bounded by 0 and 1 using the sigmoid function $\sigma$), the two key vectors $\boldsymbol{k}_1, \boldsymbol{k}_2$, and the new value $\boldsymbol{v}$.

$$[\boldsymbol{k}_1, \boldsymbol{k}_2, \boldsymbol{v}] = \phi(\boldsymbol{W}_{\text{write}}\boldsymbol{h}_t) \tag{1}$$

$$\beta = \sigma(\boldsymbol{W}_\beta \boldsymbol{h}_t) \tag{2}$$

The purpose of writing to memory is to learn a **context-specific** association between the input pattern $\boldsymbol{k}_1 \otimes \boldsymbol{k}_2$ and the output pattern $\boldsymbol{v}$. The usage of the tensor-product in the input pattern factorises the the representational space which guarantees unique orthogonal vector representations for novel key pairs. A specific example of such is given and demonstrated by Schlag & Schmidhuber (2018) where the first key learns to represent an entity and the second key a specific action, thereby, learning a representational space that generalises to never seen entity and action compositions.

In stark contrast to the complex memory operations of the TPR-RNN, we employ a single, simple, and word-level operation which is closely related to the perceptron learning rule (Rosenblatt, 1958). It allows the model to replace the previous association $\boldsymbol{v}_{\text{old}}$ with a convex combination of the old and new value $\beta\boldsymbol{v} + (1 - \beta)\boldsymbol{v}_{\text{old}}$. With the scalar $\beta$ the LSTM controls if the new association fully replaces the previous value ($\beta = 1$) or if the information of both are mixed together. Our fast weight update works as follows: First, the current value $\boldsymbol{v}_{\text{old}}$ that is associated with $\boldsymbol{k}_1 \otimes \boldsymbol{k}_2$ is retrieved. Second, we remove the old association from the map by subtracting $\text{vec}(\boldsymbol{k}_1 \otimes \boldsymbol{k}_2) \otimes \boldsymbol{v}_{\text{old}}$ from our memory, where vec vectorises the matrix. Third, we add $\text{vec}(\boldsymbol{k}_1 \otimes \boldsymbol{k}_2) \otimes (\beta\boldsymbol{v} + (1-\beta)\boldsymbol{v}_{\text{old}})$. All three steps can be achieved at once using the following update rule (see appendix section B for the proof):

$$\boldsymbol{F}'_t = \boldsymbol{F}_{t-1} + \beta \, \text{vec}(\boldsymbol{k}_1 \otimes \boldsymbol{k}_2) \otimes (\boldsymbol{v} - \boldsymbol{v}_{\text{old}}). \tag{3}$$

To prevent the fast weights from potentially growing endlessly, we scale down the fast weights whenever $||\boldsymbol{F}'_t||_2 > 1$. This is achieved through the following element-wise scaling.

$$\boldsymbol{F}_t = \frac{\boldsymbol{F}'_t}{\max(1, ||\boldsymbol{F}'_t||_2)}. \tag{4}$$

### 3.1.2 READING

For each step of the input sequence, the model queries the memory in order to retrieve a previously stored value. Due to the keys and values being generated separately, the network can retrieve values which are informationally independent from their keys. In order to perform more complex associative inference, like e.g. transitive inference ($a \rightarrow b, b \rightarrow c$, therefore, $a \rightarrow c$), we employ multiple reads where we use the retrieved value as one of the keys in the next query (see equation 7).

$$\boldsymbol{n}_t^{(0)} = \phi(\boldsymbol{W}_n \boldsymbol{h}_t) \tag{5}$$

$$\boldsymbol{e}_t^{(i)} = \phi(\boldsymbol{W}_e^{(i)} \boldsymbol{h}_t), 1 \leq i \leq N_r \tag{6}$$

$$\boldsymbol{n}_t^{(i)} = \text{LN}(\boldsymbol{F}_t \, \text{vec}(\boldsymbol{n}_t^{(i-1)} \otimes \boldsymbol{e}_t^{(i)})), 1 \leq i \leq N_r \tag{7}$$

$$\text{FWM}(\boldsymbol{h}_t, \boldsymbol{F}_t) = \boldsymbol{W}_o \boldsymbol{n}_t^{(N_r)}. \tag{8}$$

Here LN refers to layernorm without the learned element-wise affine map (Ba et al., 2016b), vec reshapes the matrix into a vector, $\phi$ is the hyperbolic tangent function, and the matrices $\boldsymbol{W}_n, \boldsymbol{W}_e^{(i)} \in \mathbb{R}^{d_{\text{FWM}} \times d_{\text{LSTM}}}, i \in \{1..N_r\}$ and $\boldsymbol{W}_o \in \mathbb{R}^{d_{\text{LSTM}} \times d_{\text{FWM}}}$ are regular slow weights trained by gradient descent which allow us to decouple the dimensionality of the LSTM from the dimensionality of the FWM. In eq. 7, $\boldsymbol{F}_t$ is the multi-linear map which we query using the LSTM-generated "input" $\boldsymbol{e}^{(i)}$ and the previous retrieval $\boldsymbol{n}^{(i-1)}$ (except for the first query where both keys are LSTM-generated).

## 4 EXPERIMENTS

### 4.1 CONCATENATED-BABI

The bAbI tasks is a popular toy dataset to benchmark neural networks with memory augmentations and reasoning capabilities (Weston et al., 2015a). It consists of a set of short stories with

questions embedded in the text. The stories were generated by simulating multiple entities in a virtual environment and cover different contexts in which entities change their state on their own or through an interaction. Each story-sample belongs to one of 20 different tasks that the authors of the dataset considered important for intelligent dialogue agents. The tasks contain questions which require reasoning capabilities like deduction, coreference, or counting. All tasks require some level of symbolic reasoning, and the first neural and non-neural baselines demonstrated poor generalisation performance on test data (Weston et al., 2015a).

We aim to improve the bAbI benchmark as a means of developing intelligent dialogue agents. To this end, we propose concatenated-bAbI (catbAbI): an infinite sequence of bAbI stories. catbAbI is generated from the bAbI dataset and during training, a random sample/story from any task is drawn without replacement and concatenated to the ongoing story. The preprocessing for catbAbI addresses several issues: it removes the supporting facts, leaves the questions embedded in the story, inserts the correct answer after the question mark, and tokenises the full sample into a single sequence of words. As such, catbAbI is designed to be trained in an autoregressive way and analogous to closed-book question answering.

catbAbI models can be trained in two different ways: language modelling mode (LM-mode) or question-answering mode (QA-mode). In LM-mode, the catbAbI models are trained like autoregressive word-level language models. In QA-mode, the catbAbI models are only trained to predict the tokens that are answers to questions—making it more similar to regular bAbI. QA-mode is simply implemented by masking out losses on non-answer predictions. In both training modes, the model performance is solely measured by its accuracy and perplexity when answering the questions. Performance on non-answers is irrelevant on catbAbI because the tokens are either very predictive or inherently unpredictable, and there is nothing appealing to be learned. Despite measuring performance only for answers, we argue that LM-mode is interesting for three reasons. First, LM-mode removes the bias of knowing which words would benefit from a symbolic inference mechanism. Second, LM-mode trains the model on a sequence with tokens which are inherently unpredictable. Such tokens could also appear in natural language and might harm the model's ability to learn a useful representation of the story. Indeed, in the next section, we will give evidence for such a generalisation gap. Third, the LM-mode setting allows us to directly compare our method with state-of-the-art language models.

### 4.1.1 RESULTS

We compare our FWM directly with the current state-of-the-art on word-level bAbI: Metalearned Neural Memory (MNM; Munkhdalai et al. (2019)). We also include two strong autoregressive word-level language models as baselines: a regularized LSTM (Merity et al., 2018; Melis et al., 2017) and a regularized Transformer-XL (TXL; Dai et al. (2019)). Lastly, we also evaluate Ba's Fast Weights which attend to the recent past (JBFW; Ba et al. (2016a)) but were unable to find hyperparameters that converged. We truncate backpropagation through time (tBPTT) to 200 tokens for all models and limited the amount of GPU memory to ~16GB for practical reasons. For every model, we performed a hyperparameter search in QA mode over the first 3k steps of which a smaller selection was trained for 30-60k steps. For all models, we adopt the best QA mode hyperparameters for the LM mode results. Table 1 lists the best accuracy and perplexity of each model over three seeds while figure 2 shows the learning curves of the best seeds. Further hyperparameter search results can be found in the appendix section F.

Our experiments on catbAbI show that a regularized, 4-layer deep, and residual LSTM, and a 3-layer deep TXL with attention over the last 1400 tokens, achieve strong performance on catbAbI. MNM, on the other hand, suffered a ~10% drop in QA mode accuracy compared to its performance on bAbI which demonstrates the increased difficulty of catbAbI. The JBFW model is not able to make meaningful predictions on catbAbI which may be due to its inability of removing previous associations and fixed fast weight memory decay. Our FWM achieves an excellent accuracy on catbAbI while being by far the smallest in parameter count and weight to activation ratio. The performance gap between FWM and MNM suggests the importance of our fast weight memory mechanism. In figure 3 we visualise how the FWM can chain memories from different points in time to perform transitive inference.

We chose to include the TXL model in our comparison due to its autoregressive nature and strong performance in large-scale language modelling benchmarks. However, we point out that the TXLs

Table 1: Accuracy and perplexity on test data over three seeds of each model's best hyperparameters setting according to our hyperparameter search. Detailed hyperparameters and results can be found in the appendix section F.

| Mode | JBFW | LSTM | TXL | MNM | FWM |
|---|---|---|---|---|---|
| QA acc | $13.22 \pm 0.0$ | $80.88\% \pm 0.30$ | $87.66\% \pm 2.82$ | $88.97\% \pm 6.28$ | **96.75%** $\pm$ **0.05** |
| QA ppl | $31.19 \pm 8.8$ | $1.93 \pm 0.11$ | $1.50 \pm 0.14$ | $2.50 \pm 1.07$ | $1.36 \pm 0.06$ |
| LM acc | $0.0 \pm 0.0$ | $80.15\% \pm 0.40$ | $90.23\% \pm 1.01$ | $69.30 \% \pm 16.60$ | **93.04%** $\pm$ **0.62** |
| LM ppl | $160.3 \pm 24.3$ | $1.84 \pm 0.02$ | $1.39 \pm 0.03$ | $2.60 \pm 1.02$ | $1.45 \pm 0.14$ |
| weights | 548k[3] | 8M | 10.5M | 1.1M | 694k |
| activations | 263k | 4096 | 4.3M[4] | 30.5k | 33.3k |

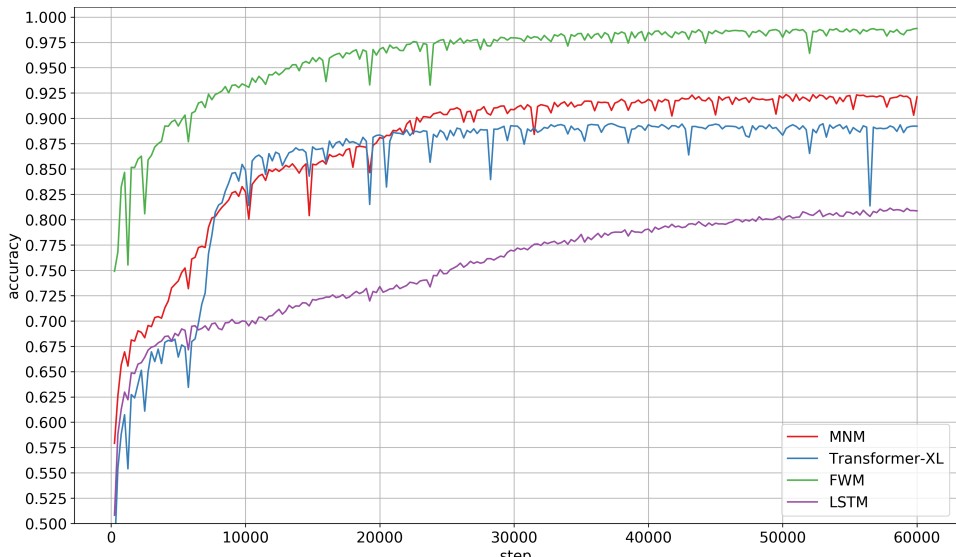

Figure 2: QM model validation accuracy of the best-over-all seeds of each model over training steps.

context window is larger than the average bAbI story. In this case, due to the shortness of the stories, catbAbI becomes more of an open-book problem for the TXL model since it has the capability of looking up representations of its previous input whereas the RNN models do not. This fundamentally limits the TXL model as it can only condition its prediction on information that is no longer than its attention window to past states. The RNN models, which are general function approximators, for better or for worse, are instead forced to learn to carry the necessary information through time.

## 4.2 META-REINFORCEMENT LEARNING

Meta reinforcement learning (Meta-RL) applies meta-learning (Schmidhuber, 1987; Hochreiter et al., 2001; Finn et al., 2017) to the field of reinforcement learning (Schmidhuber, 1994). An agent is trained on multiple environments (or tasks) and receives environmental feedback as part of its input. To maximise its total reward in an environment, the agent has to leverage the feedback signals and adapt. A successful agent is capable of maximising its reward in novel environments that it has not been exposed to during training. Recent work achieved notable progress in this domain (Santoro et al., 2016; Mishra et al., 2018; Kirsch et al., 2020). We experiment with tasks drawn randomly from a large set of partially observable Markov decision processes (POMDPs). In this set, every environment consists of precisely five states and three actions. Globally, every environment can be viewed as a sparse directed graph where nodes are locations, and the directed edges are one-way modes of transportation—similar to a metro transit map of a city (Graves et al., 2016). To generate

---

[3]Bigger JBFW models did not improve performance. See appendix section F.5.

[4]The number of immutable activations is $512 \times 2 \times 3 \times (1200 + 199)$ while the number of mutable activations is merely $512 \times 2 \times 3 = 3072$. Only the TXL model maintains immutable activations.

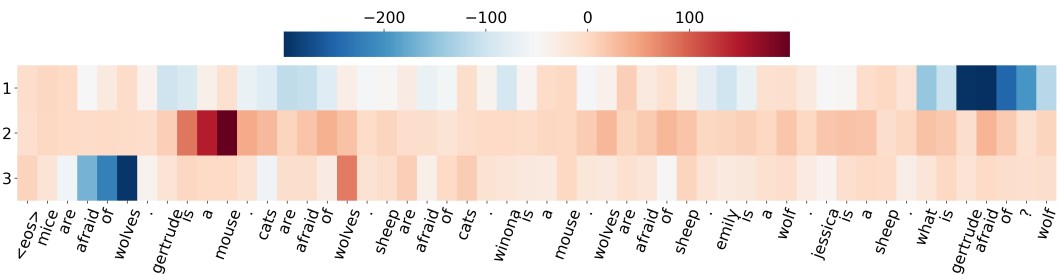

Figure 3: A visualisation of the FWMs ability to chain independent associations to perform transitive reasoning on the catbAbI validation data. The colour of each grid cells represent the dot product $\langle \boldsymbol{k}_1 \otimes \boldsymbol{k}_2, \boldsymbol{n} \otimes \boldsymbol{e} \rangle$ where $\boldsymbol{k}_1, \boldsymbol{k}_2$ are the write keys of each previous position while $\boldsymbol{n}, \boldsymbol{e}$ refers to the respective queries generated at "?" (second position from the right) for each of the $N_r = 3$ memory reads. The first query matches most with the keys at the recent positions where the input was "gertrude" and "afraid" (first row of grid cells). The second query, which partially consists of the value retrieved from the first query, matches with the "getrude is a mouse" section. The third query, which partially consists of the value retrieved from the second query, matches with the "mice are afraid of wolves" section. Finally, the FWM correctly outputs the next word and answer to the question: wolf (not seen). This likely completes the deduction: gertrude is a mouse, mice are afraid of wolves, therefore, gertrude is afraid of wolves.

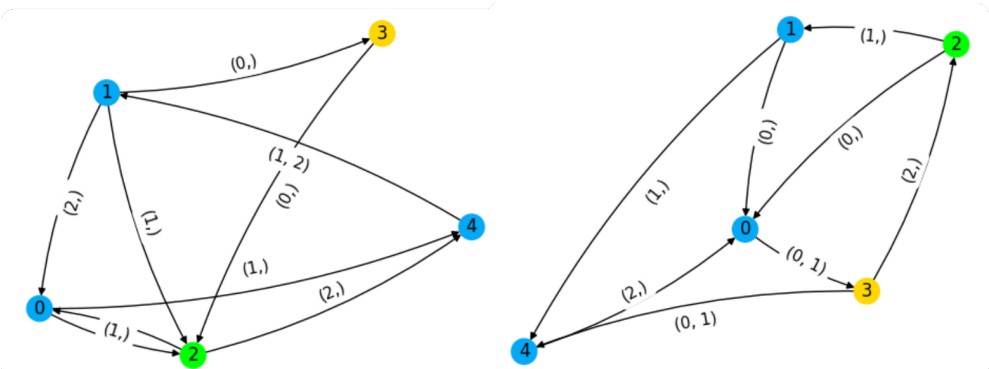

Figure 4: Two randomly generated environments with the agent's location coloured in green and the reward location coloured in yellow. Edge labels indicate the set of valid actions (0, 1, or 2) to transition along that arrow. Invalid actions are not visualised. The graph and the locations of the agent and reward are set randomly at the beginning of the experiment. If the agent reaches the reward location or did not reach it after six steps, both are randomly reset.

a new environment, we sample the adjacency matrix of the graph such that actions are deterministic, and every location is reachable from any other location (see figure 4). We sample graphs such that there are no actions that lead to the same location, and such that *not* every action is always a valid way of transitioning. We added the exact algorithm to generate graphs, as well as further details, to the appendix section I.

The agent's goal is to reach the reward location. Upon arrival, the agent receives the reward, followed by a random reset of the agent's and reward's location. Whenever the agent takes an action that does not lead to a new location, it receives a penalty. At every step, the agent receives as an input: its current location, the reward location, its last action, and the reward received so far.

We run our experiment for 30 steps and compare our FWM to an LSTM baseline. Both methods are trained on the same training set of 600 graphs and tested on 600 novel graphs. We optimise our agent with the Advantage Actor-Critic (A2C) algorithm, a non-asynchronous version of the A3C method (Mnih et al., 2016). In our experiments, the LSTM-based agent requires more episodes, a bigger network, and eventually overfit to the training graphs. The FWM-based agent however trains faster and generalises to randomly sampled graphs.

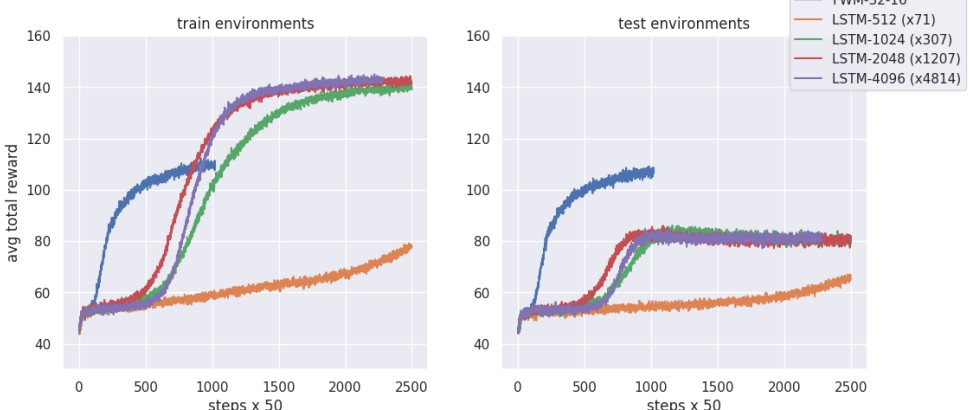

Figure 5: Average total reward of the agent when trained on 600 random graphs (left plot) and tested on 600 different graphs (right plot). The FWM agent (blue) has a slow LSTM with 32 hidden units and a fast weight memory of size $16 \times 16^2$. We compare to LSTM agents with different sized hidden states. The largest LSTM has 4096 hidden units (red) which roughly matches the number of temporal variables of the FWM. The FWM has 14k trainable weights which is by far the lowest. The largest LSTM has 67.4M weights which is roughly 4814 times more than the FWM. The relative factor of each LSTM is added to the legend. All LSTMs take longer to train and eventually overfit on the training data. Due to the overfitting, the LSTM does not have to explore, which results in a higher total reward on training environments but a lower total reward on test environments.

We argue that the bAbI stories and the episodes on the graphs are similar in the following three ways. First, in both problems, the network has to construct a useful and *context-specific* representation from its ongoing input. Second, as part of its input, the network repeatedly receives an objective (the reward location versus the question) which requires the exploitation of the context-specific information. Third, the model has to produce a discrete sequence (actions in the environment in RL and reasoning steps in catbAbI) to optimise its training signal (high reward versus low uncertainty).

## 4.3 LANGUAGE MODELLING

Comparing FWM to autoregressive language models on catbAbI begs the question: how does FWM perform on popular word-level language modelling datasets such as Penn Treebank (PTB; Mikolov et al. (2010)) or WikiText-2 (WT2; Merity et al. (2017))? It is unclear to which extend a symbolic inference mechanism is beneficial for language modelling. PTB and WT2 contain virtually no questions and are constructed from Wikipedia and news articles which are designed to be easily parsed by the reader. Nevertheless, in figure 6 we show how our FWM exploits recurring subject names to reduce its uncertainty. Not many memory augmented NNs have been able to bridge from small and toy reasoning tasks to general language models—and those which did, underperformed (Paperno et al., 2016; Sukhbaatar et al., 2015). We use the regularized 3-layer AWD-LSTM (Merity et al.,

Table 2: Best perplexity on the test data of Penn Treebank (PTB) and WikiText-2 (WT2) from three seeds. Detailed results can be found in the appendix in table 5. All PTB models have roughly 24M parameters and all WT2 models have roughly 37M parameters. The AWD-TXL is the Transformer-XL architecture as reported by Dai et al. (2019) with the necessary AWD-style regularisation, model averaging, and softmax temperature tuning (see appendix section H).

| Model | PTB | | WT2 | |
| | Validation | Test | Validation | Test |
|---|---|---|---|---|
| AWD-LSTM (Merity et al., 2018) | 60.0 | 57.3 | 68.6 | 65.8 |
| AWD-TXL (Dai et al., 2019) | - | 54.52 | - | - |
| AWD-TXL (ours) | 59.39 | 56.50 | 65.73 | 63.11 |
| AWD-FWM (ours) | 56.76 | **54.48** | 63.98 | **61.65** |

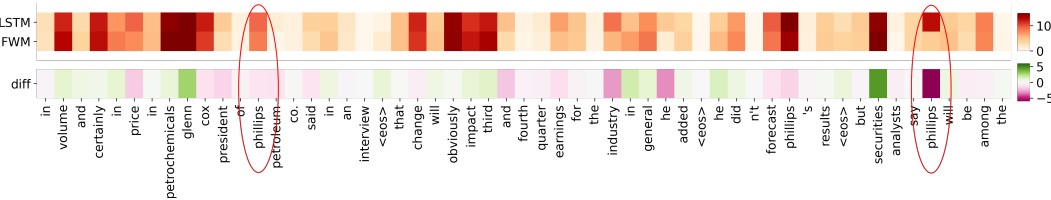

Figure 6: Loss comparison between the LSTM and our FWM on a section of the PTB test set. The colour of the grid cells in the first row stands for the cross-entropy error of the LSTM and FWM model. The second row, for their respective difference. Our FWM sometimes shows a lower error on rare subject words such as names of companies and people once they have been introduced. As seen in the red circles, the initial mentioning of "phillips" has similar uncertainty between the LSTM and FWM but shortly after that the subject of the sentences is more predictable and the FWM is more certain (4.3 bits difference) whereas the LSTM's uncertainty remains roughly on the same level (12.8 bits).

2018) as the slow RNN in our FWM model to minimize further hyperparameter search. The experimental results in table 2 demonstrate a relative improvement over the AWD-LSTM baselines, which suggest the benefit of our FWM even in language modelling benchmarks. However, in contrast to catbAbI, all three models achieve very similar results which might indicate that PTB and WT2 do not benefit as strongly from an associative reasoning capacity. We added the experimental details to the appendix section H.

Since the publication of AWD-LSTM (Merity et al., 2018), various extensions (some of which are orthogonal to our memory augmentation) have been proposed (Krause et al., 2018; Merity et al., 2018; Yang et al., 2018). In this work, we are not primarily interested in beating the state-of-the-art in language modelling and leave it for future work to explore the possible synergies between these methods.

## 5 DISCUSSION

An order-three memory tensor is a computationally demanding method for constructing compositional state representations. With vector components in $\mathbb{R}^n$, the tensor product computation alone has a space and time complexity of $O(n^3)$. For practical reasons, this forces the FWM to remain small, relative to the slow NN, which limits the number of associations that can be maintained at once. Previous work has proposed approximations of such memory tensors in a variance-optimal way (Schlag et al., 2019). In our ablation experiments in section E, we show on catbAbI that concatenating the keys results in a performance accuracy drop of ~5%. We also experiment with fewer read operations (smaller $N_r$) which also results in a performance degradation (appendix figure 7). However, further improvements might not come from scaling up but from more general symbolic manipulations. We address the capacity of the FWM and the necessity of the tensor product from a linear hetero-associative memory perspective in section A of the appendix. Finally, our fast weight memory can be thought of as a primitive "working memory" of the model—analogous to the working memory in the human brain (Spalding et al., 2018). This idea is supported by recent work which proposes a cognitive model of the human brain that is based on such higher-order tensors (Tresp & Ma, 2017).

## 6 CONCLUSION

Our new FWM is a fast weights architecture capable of learning from synthetic data to answer questions which require various symbolic reasoning skills. To improve generality, we overcome issues of the popular bAbI dataset by introducing more general and more difficult variation dubbed catbAbI. We report excellent performance on catbAbI and compare with strong baselines based on state-of-the-art language models, as well as, the previous state-of-the-art in word-level bAbI. We also apply the FWM in a challenging meta-reinforcement learning environment where the agent generalises to novel environments by learning from its observations and actions. Finally, in a self-supervised setting, we apply the FWM to word-level language modelling on PTB and WT2 where it beats the AWD-LSTM and AWD-Transformer-XL baselines.

## ACKNOWLEDGEMENTS

We thank NVIDIA Corporation for donating several DGX machines, and IBM for donating a Minsky machine. This research was supported by an European Research Council Advanced Grant (no: 742870).

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

CONTENTS (APPENDIX)

## A    FURTHER DISCUSSION

One way of assessing the capacity of the third-order tensor memory is its rank (which is analogous to the rank of a matrix). However, there exists no general algorithm to determine the rank of a given higher-order tensor $\mathbf{A} \in \mathbb{R}^{I \times J \times K}$. There exists only a loose upper bound described by $\text{rank}(\mathbf{A}) \leq \min\{IJ, IK, JK\}$ (Kruskal, 1989; Kolda & Bader, 2009).

It might be tempting to simplify the FWM by replacing the outer-product of the input with a concatenation as a means to reduce the space and time complexity. However, in highly compositional domains, the concatenated input will suffer from interference between memories. Consider a problem which, from a set of 10 symbols, requires the association of any three symbols represented by the vectors $s, r, t \in \mathbb{R}^{10}$. In the case of a concatenation, one rank of the fast weight memory is $[s; r] \otimes t$ where we refer to $[s; r]$ as the key representation. The read vectors $s', r' \in \mathbb{R}^{10}$, are then concatenated and matrix multiplied to retrieve the previous association $\hat{t} = F[s'; r']$. Here we refer to $[s'; r']$ as the query representation. Since there are ten distinct symbols of which any two can behave as a key representation, there exist $10^2 = 100$ unique key patterns. To guarantee noise-free retrieval in any context, the vectors of the key representations have to be orthogonal. However, $[s'; r']$ is only a 20 dimensional space which means that certain key representations cannot be used simultaneously without interference. The tensor product, on the other hand, is capable of noise-free retrieval because it represents the key as $s \otimes r \in \mathbb{R}^{10 \times 10}$ which allows for 100 orthogonal keys and as such the possibility of noise-free retrieval. We conclude that if the problem is highly compositional, in a sense that every component can be composed with any other component, then the tensor product will be better suited than a concatenation. Experimentally we evaluate concatenated keys in section E. The results show that concatenated keys will result in a slightly worse performance (see figure 8). As an alternative, a non-linear memory, e.g. through the use of a softmax, would not require orthogonality in it's keys to be free of interference and could result in a larger storage capacity.

## B    DERIVATION OF THE UPDATE RULE

**Theorem B.1.** *Given two key vectors $k_1, k_2 \in \mathbb{R}^d$ and two value vectors $v_{old}, v_{new} \in \mathbb{R}^d$ with $d \in \mathbb{Z}_{>0}$, a mixing coefficient $\beta \in (0, 1)$, and a fast weight memory $F_{old} = \text{vec}(k_1 \otimes k_2) \otimes v_{old}$ where vec refers to the vectorisation of the higher-order tensor, then the (recurrent) fast weight update rule given by $F_{old} + \beta \, \text{vec}(k_1 \otimes k_2) \otimes (v_{new} - v_{old})$ results in $F_{new} = \text{vec}(k_1 \otimes k_2) \otimes [(1 - \beta)v_{old} + \beta v_{new}]$.*

*Proof.*

$$F_{\text{new}} = F_{\text{old}} + \beta \, \text{vec}(k_1 \otimes k_2) \otimes (v_{\text{new}} - v_{\text{old}}) \tag{9}$$
$$= \text{vec}(k_1 \otimes k_2) \otimes v_{\text{old}} + \text{vec}(k_1 \otimes k_2) \otimes (\beta v_{\text{new}} - \beta v_{\text{old}}) \tag{10}$$
$$= \text{vec}(k_1 \otimes k_2) \otimes [v_{\text{old}} + \beta v_{\text{new}} - \beta v_{\text{old}}] \tag{11}$$
$$= \text{vec}(k_1 \otimes k_2) \otimes [(1 - \beta)v_{\text{old}} + \beta v_{\text{new}}] \tag{12}$$

$\square$

## C    A COMMENT ON THE REGULAR BABI DATASET AND PREVIOUS WORK

The bAbI tasks is a popular toy dataset to benchmark neural networks with memory augmentations and reasoning capabilities (Weston et al., 2015a). It consists of a set of short stories with questions embedded in the text. The stories were generated by simulating multiple entities in a virtual environment and cover different contexts in which entities change their state or interact with each other. Each story-sample belongs to one of 20 different tasks that the authors of the dataset considered important for intelligent dialogue agents. The tasks contain questions which require reasoning capabilities like deduction, coreference, or counting. All tasks require some level of symbolic reasoning, and the first neural and non-neural baselines demonstrated poor generalisation performance on test data (Weston et al., 2015a). In addition to the story sentences, the questions, and the answers, the dataset also included supporting facts which demarcated question-relevant sentences in the story. The stories often follow multiple parallel plots where each new sentence is advancing one of the plots by a single fact.

The bAbI dataset did not include a strict experimental protocol which resulted in several variations that differed slightly. Early methods achieved good results by relying on the supporting facts (Weston et al., 2015b; Kumar et al., 2016) or other supervised training signals (see e.g. Johnson (2017); Li et al. (2016)).

Some researchers achieved great results by reformatting the data such that the question is read before the story or, similarly, by giving the model the capacity to lookup parts of the story, e.g. through some attentional mechanism, after the question has been read (Sukhbaatar et al., 2015; Xiong et al., 2016; Dehghani et al., 2019). Such methods have shown to be useful for answering questions while maintaining access to the full story. We argue that this is similar to open-book question answering. In such a setting, the model is incentivised to look up information instead of capturing the useful bits of the data it has seen. The advantage of the latter becomes more evident in a different scenario: imagine the model is processing a book where a user can ask a question about the content at any time. An open-book approach will have to store all previous sentences in its memory and apply its answer-search mechanism to all of the data. Instead, a closed-book approach would store a compressed version of the story, or the question-relevant information of the story.

It is essential to acknowledge that the sentences in the bAbI stories of all tasks are short and simplistic. Virtually every sentence contains precisely one fact. Because of that, it might be that sentence-level models have an advantage over word-level models. Indeed, a previous sentence-level model has reported poor performance in the word-level setting (Schlag & Schmidhuber, 2018). This limits their generality since sentences in natural language are often not limited to a single fact.

Lastly, even though the bAbI dataset was initially designed with the questions embedded in the story, virtually all methods so far preprocess the dataset such that a sample with four questions is split into four samples with one question each (Weston et al., 2015b). This arguably simplifies the problem because the model does not need to maintain the state of other entities which are not relevant to the question once it is read. However, it remains to be tested if this would result in inferior performance.

## D  CONCATENATED-BABI DETAILS

Concatenated-bAbI (catbAbI) is a preprocessing and experimental procedure to evaluate autoregressive models in their capability of predicting words which require certain reasoning skills (here answers of questions). In this work we only focused on the 10k samples per task version of bAbI but all our scripts can be applied to the 1k version as well. We used the same train/test/valid split of the data as in regular bAbI. In contrast to previous work, we do not split the stories to contain only one question. We remove the sentence indecies and concatenate the sentences with answers following a question mark into one long sequence of words. The preprocessed data is a shuffled list of samples. Each sample comes with its task id for diagnosis. All answers are preceeded by a question mark.

To ensure that stories do not overlap and become ambiguous, we add a special end-of-story token before concatenating the new story. For each word, the preprocessing script provides its task id to measure the performance on different tasks. Similarly, it also provides a special answer token which signifies if the current word is an answer or not. Naturally, the task id and answer information are not provided to the model as an input. The validation and test data are processed likewise, but for a proper comparison of various models, validation and test data are shuffled only once[5]. During training and evaluation, the validation and test stories are drawn deterministically.

Table 3: Statistics of the catbAbI dataset based on our preprocessing of the regular bAbI data.

| subset | number of tokens | number of stories | number of questions |
|--------|------------------|-------------------|---------------------|
| train  | ~5M              | 56,376            | 179,909             |
| valid  | ~560k            | 6,245             | 19,907              |
| test   | ~560k            | 6,247             | 19,910              |

During training we uniformly sample stories without replacement and concatenate them into a long sequence. Since a question mark is not always the end of a story we resolve any ambiguity by

---

[5]We provide the preprocessed catbAbI data together with our code so future work can compare using the same validation and test sequence.

separating the stories with a special end-of-story token. The model is trained on this long sequence in an autoregressive way with truncated backpropagation. At the end of the epoch, we fill the batch with padding symbols if the sequences in the batch have different lengths.

In LM-mode we mask padding tokens and in QA-mode we mask everything except the steps with a question mark as input. At the end of the epoch we carry over the hidden states to the new epoch. Resetting all hidden states to the same or to zeros had a weak negative effect on final performance but was not explored thouroghly. For evaluation on valid and test splits a copy of the hidden state of the first batch element is used. Evaluation on valid is done throughout training with a large batch-size to maintain speed. Evaluation on test is done with a batch-size of one. During evaluation on valid and test the samples are picked sequentially to ensure that all models are evaluated on the same valid and test sequence of bAbI stories.

# E   ABLATION

We evaluate the FWM model with different number of recurrent steps. Experiments in figure 7 indicate that just one step is already achieving over 95% accuracy but more inference steps help on rarer but harder tasks. We also test a FWM version where the read and query keys are concatenated instead of multiplied through the tensor product. In this version, the FWM results in a weight matrix with $\mathbb{R}^{2d_{\text{FWM}} \times d_{\text{FWM}}}$ instead of $\mathbb{R}^{d_{\text{FWM}}^2 \times d_{\text{FWM}}}$. The results in figure 8 indicate a drop in performance.

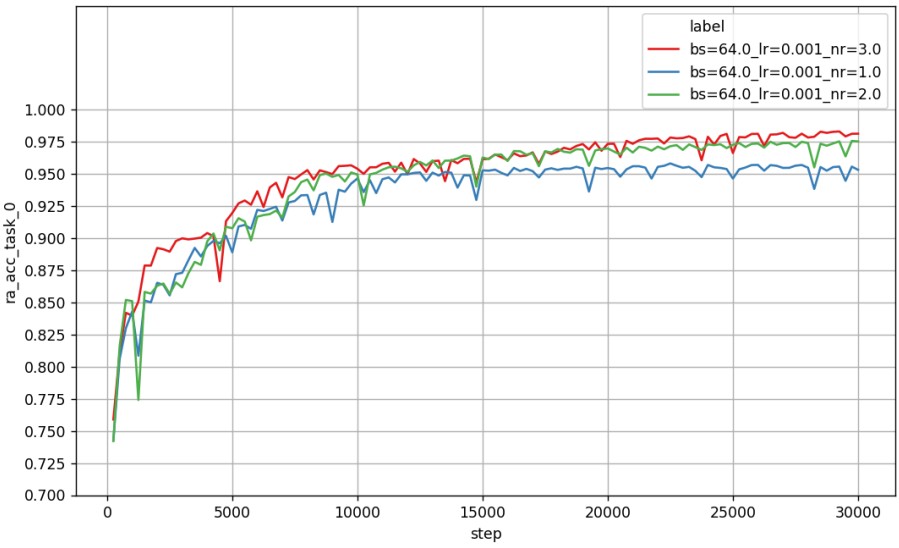

Figure 7: Comparison of the FWM with the same seed but with different $N_r$.

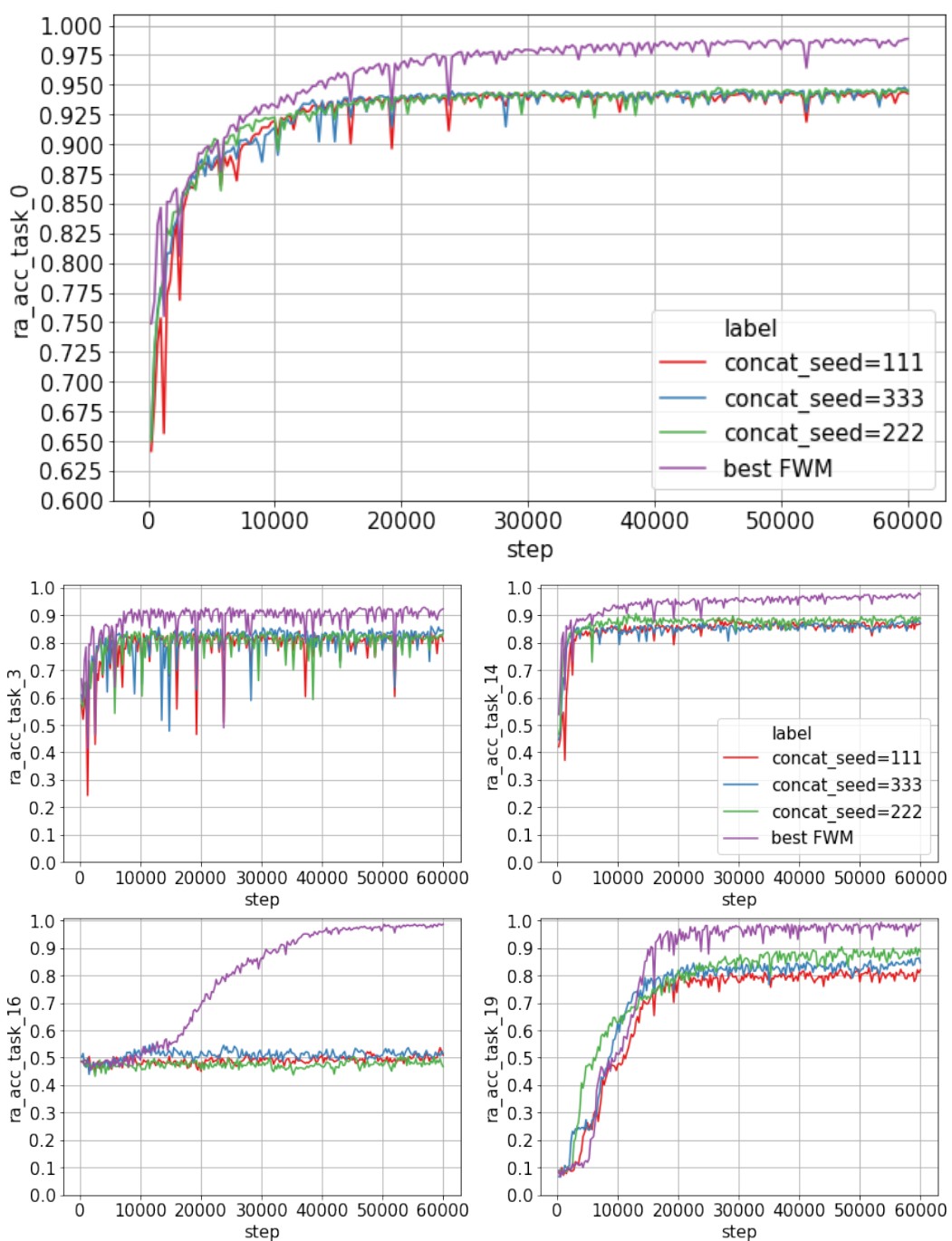

Figure 8: FWM model with a concatenated keys compared with the tensor product of the keys. With a concatenation of the respective keys and queries the Fast Weight tensor has a squared space and compute complexity $O(d_{\mathrm{FWM}}^2)$ but performs worse on average (top figure). The performance difference is limited to more complex tasks such as 3, 14, 16, 19 (bottom figures).

## F  HYPERPARAMETER SEARCH FOR CATBABI

Since catbAbI is an ongoing sequence of stories, backpropagation through time (BPTT) is infeasable
for all models which is why we truncate BPTT to the last 200 tokens. Hyperparameters were chosen
such that they fit roughly on one GPU with 16GB of memory. All models use a token embedding
size of 256 and the Adam optimizer. We exclusively tuned the hyperparameters for the QM setting
and transfer only the best to the LM setting. We run a grid search over the batch-size, learning rate,
and various model specific parameters such as dropout rates or number of layers on top of additional
manually chosen settings. For computational reasons we run two rounds of grid-search: an initial
round of 3,000 steps of which the best are moved to the second round where we train them for 30,000
or 60,000 steps. In the following subsections we give further details for each model seperately.

### F.1  FAST WEIGHT MEMORY

We set $d_{LSTM} = 256, d_{FWM} = 32, N_r = 3$ and searched experimented with two seeds for batch sizes
64, 128 and learning rates 0.0001, 0.00025, 0.0005, 0.001, 0.002.

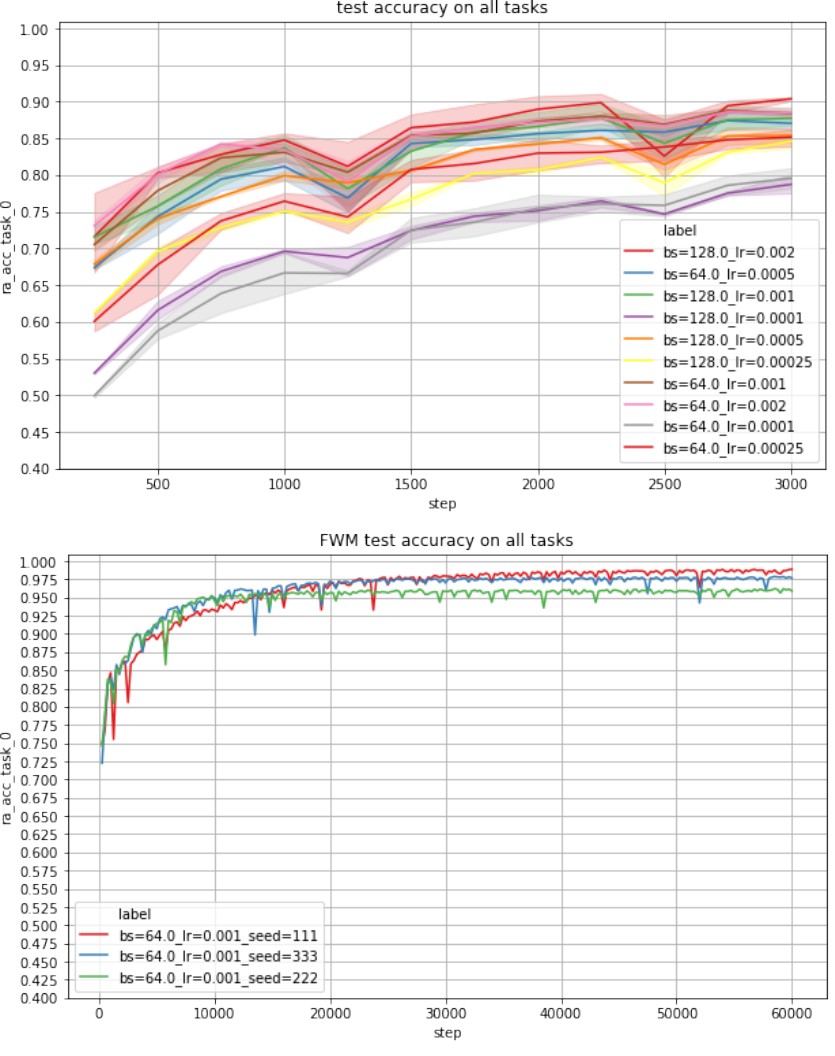

Figure 9: Top: Hyperparameter search runs for different batch sizes and learning rates of the FWM
model in the QM setting with the average accuracy on all tasks. Bottom: FWM performance over
60,000 steps with three seeds.

### F.2 METALEARNED NEURAL MEMORY

We only experimented with the plastic version of MNM as it was reported to be the best. We used the same hyperparameters for the fast weights as reported by Munkhdalai et al. (2019): 3 layer of fast weights with a dimensionality of 100. We searched over the batch sizes 64, 128; learning rates 0.00025, 0.0005, 0.001, 0.002; and meta-objective coefficient (reg) 1.0, 2.0. In the first 3,000 steps the MNM didn't show any instability but for longer runs the MNM would sometimes result in NaNs or become unstable.

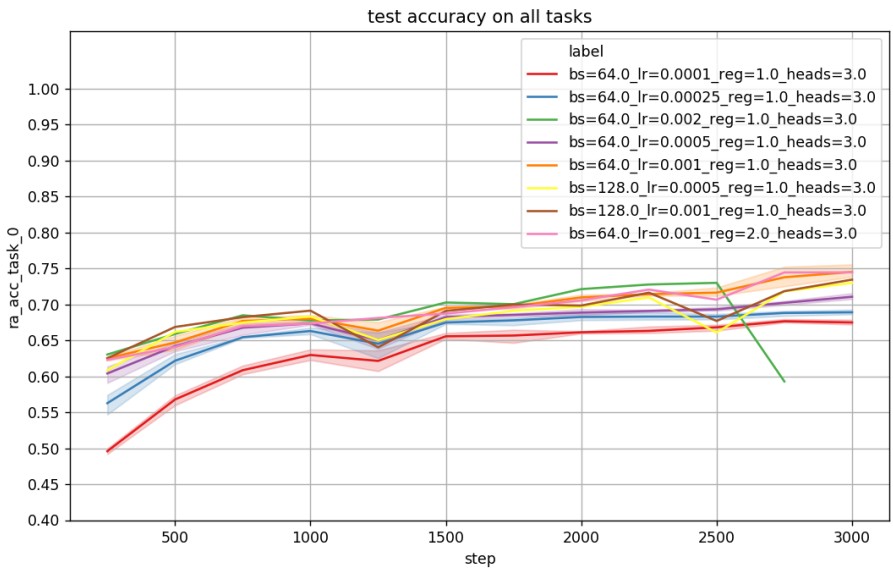

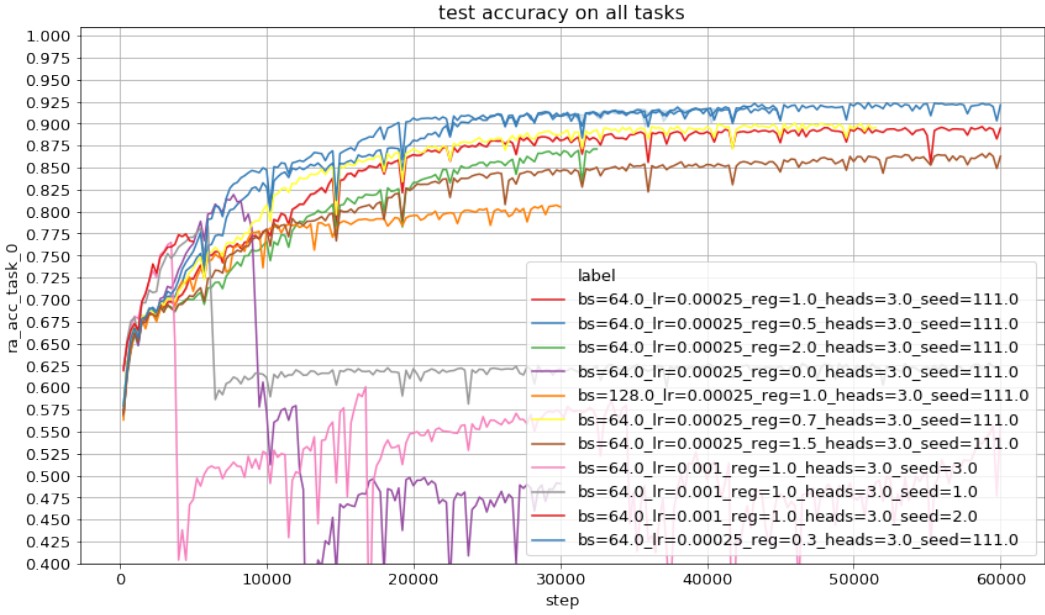

Figure 10: Top: Hyperparameter search runs for different batch sizes and learning rates of the MNM model in the QM setting with the average accuracy on all tasks. Bottom: MNM model with three different seeds, batch size 64, and learning rate 0.001 in the QM setting. Reported accuracy is the average on all tasks.

### F.3 TRANSFORMER-XL

We ported the official Transformer-XL implementation[6] to our own codebase; fully reusing the model code for our catbAbI experiments. We employ a linear learning-rate warm-up schedule over the first 1000 steps and run a grid search over batch size, learning rate, number of layers, and memory length with some additional manual selected parameters. Our best setting uses a learning rate of 0.00025, memory width of 1200, a hidden state size of $d_{\text{model}} = 512$, an inner dimension of the fully connected part of $d_{\text{inner}} = 2048$, and 3 transformer layers. Several long runs can be seen in figure 12. Our experiments show how various seeds eventually become unstable and overfit. Some settings also resulted in NaNs which we have removed from figure 12. The best performing models and most stable where 3 layer models with a large memory and a small learning rate (see figure 13).

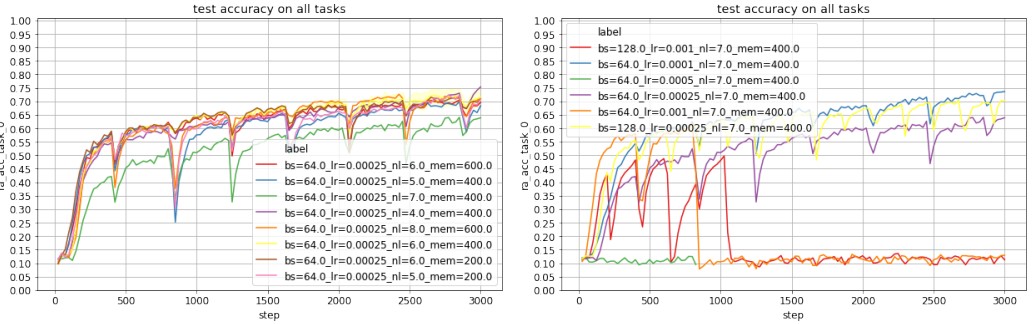

Figure 11: Hyperparameter search runs for different batch sizes and learning rates of the Transformer-XL in the QM setting with the average accuracy on all tasks. Left graph varies number of layers and memory length. Right graph varies batch size and learning rate for 7 layers.

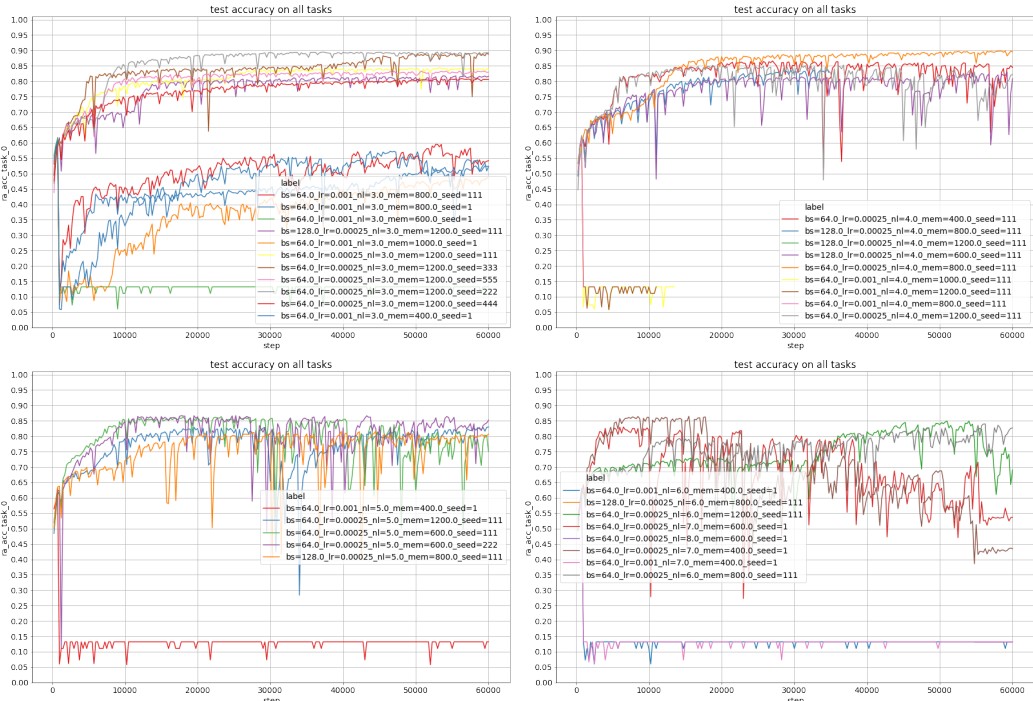

Figure 12: Long hyperparameter search runs for TXL with various layers and memory sizes. The experiments are grouped based on the number of layers. Many runs begin to diverge late into the training process.

---

[6]Source: `github.com/kimiyoung/transformer-xl/blob/master/pytorch/mem_transformer.py`

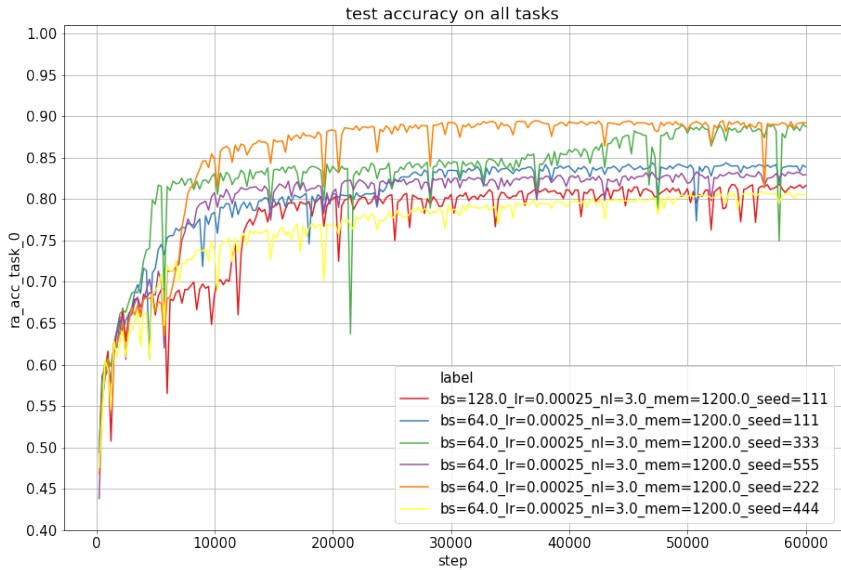

Figure 13: Various seeds for the best Transformer-XL hyperparameters: 3-layers, memory windows of 1200 tokens, a learning rate of 0.00025, and a batch size of 64.

## F.4 LSTM

We heavily regularize a four-layer stack of residually connected LSTM cells, each with 512 hidden units. Inspired by AWD-LSTM (Merity et al., 2018), we use dropout in four different ways to regularize the model. We dropout the tokens of the input sequence, elements of the embedding vector, elements of the recurrent weight matrix, and elements of the of the hidden representation between LSTM layers.

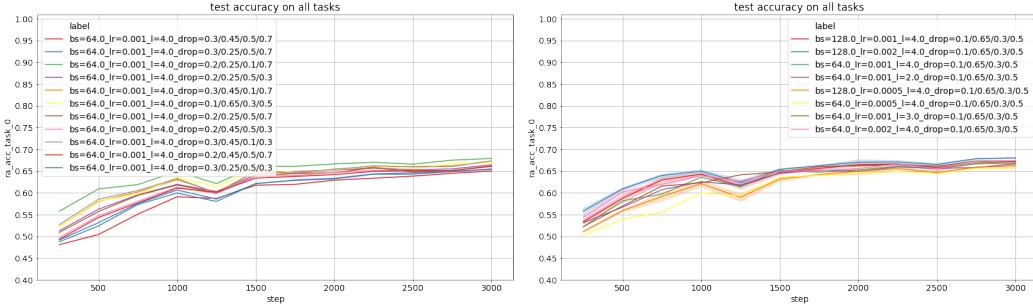

Figure 14: Hyperparameter search runs for different batch sizes and learning rates of the LSTM in the QM setting with the average accuracy on all tasks.

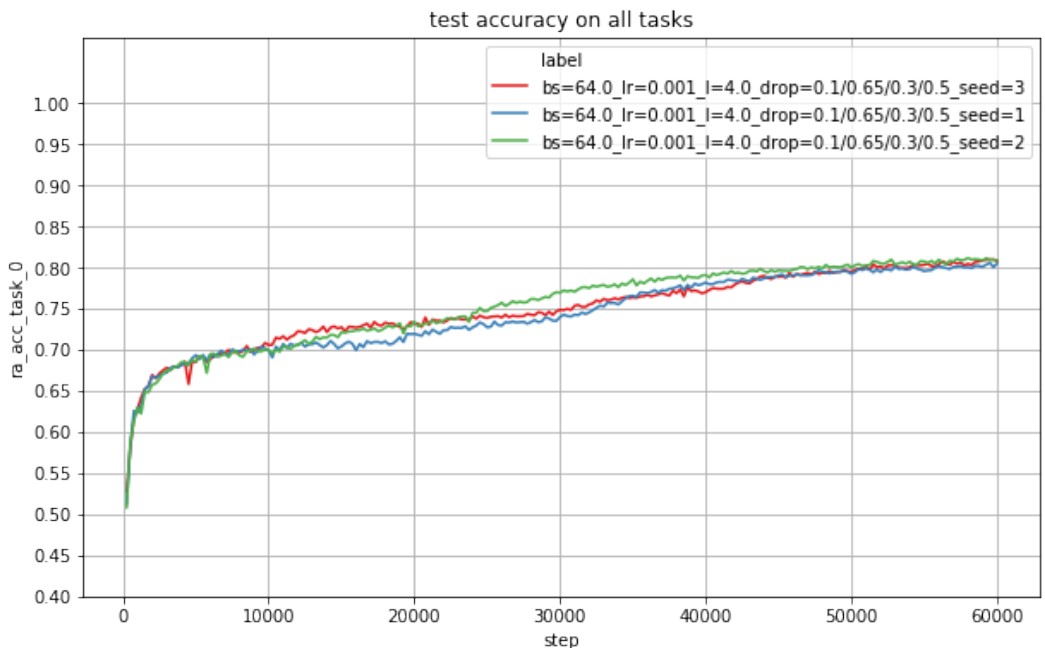

Figure 15: Average accuracy of three seeds of the best LSTM settings over all tasks on the catbAbI QM-mode dataset.

### F.5 ATTENTION TO THE RECENT PAST FAST WEIGHTS

We evaluate our own implementation of Fast Weights as introduced by Ba et al. (2016a). They propose an RNN augmented with fast weights which modulate the slow weights of an Elman RNN using a fixed fast weight learning and decay rate (JBFW). Our hyperparameter search did not result in any model performing over 15% on the test data.

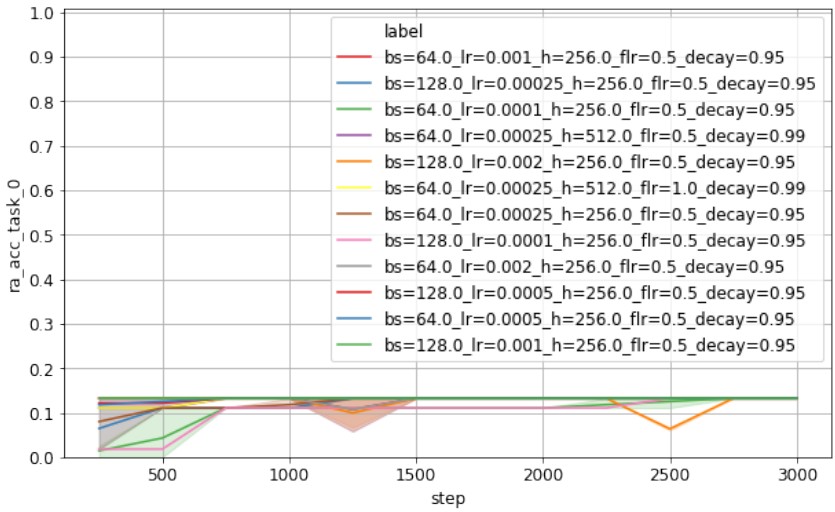

Figure 16: Hyperparameter search for the Fast Weights attending to the recent past by Ba et al. (2016a).

# G  BEST CATBABI RUNS BROKEN DOWN BY TASK

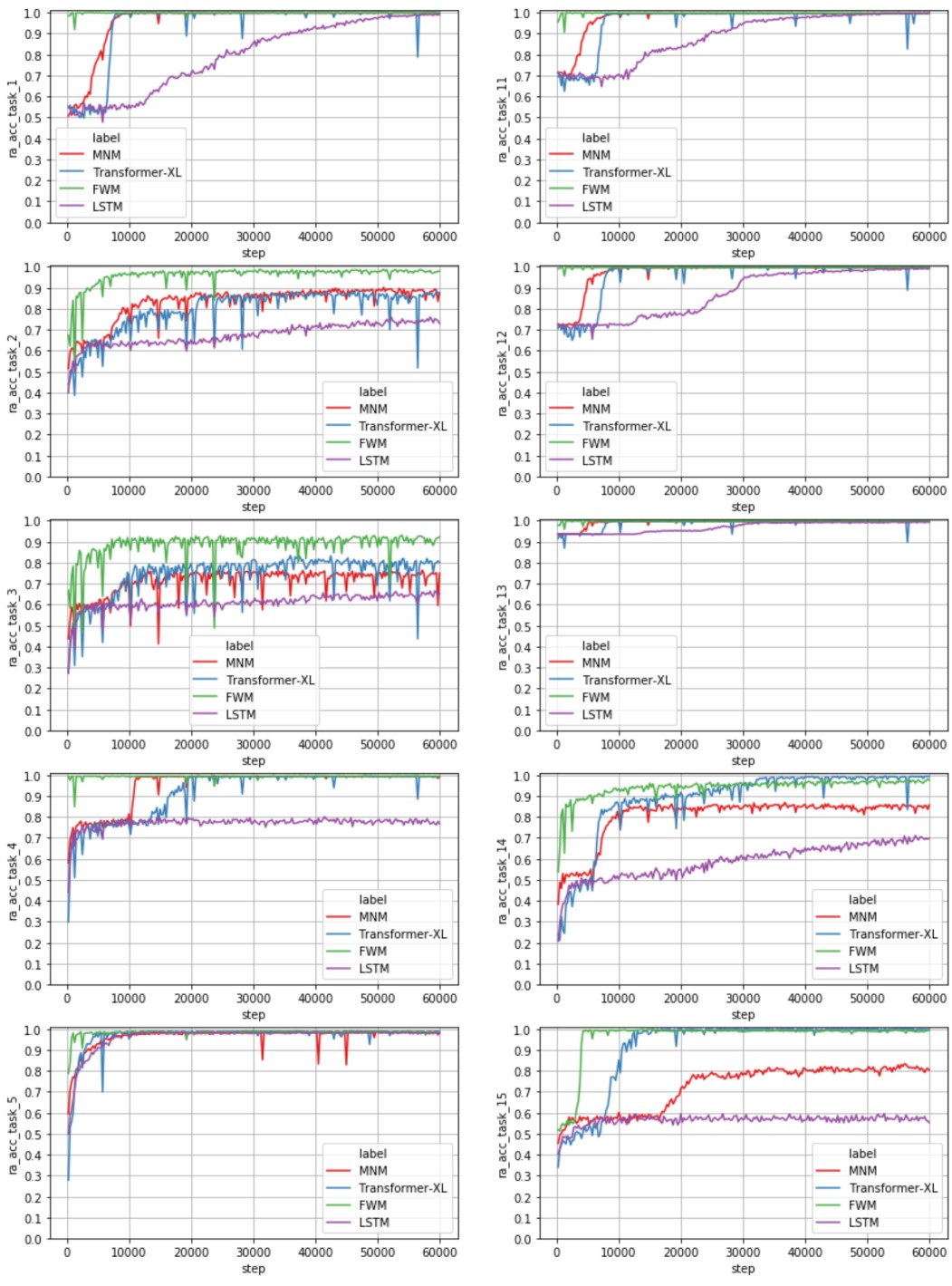

Figure 17: Per-task test set performance comparison of the best catbAbI runs (first part).

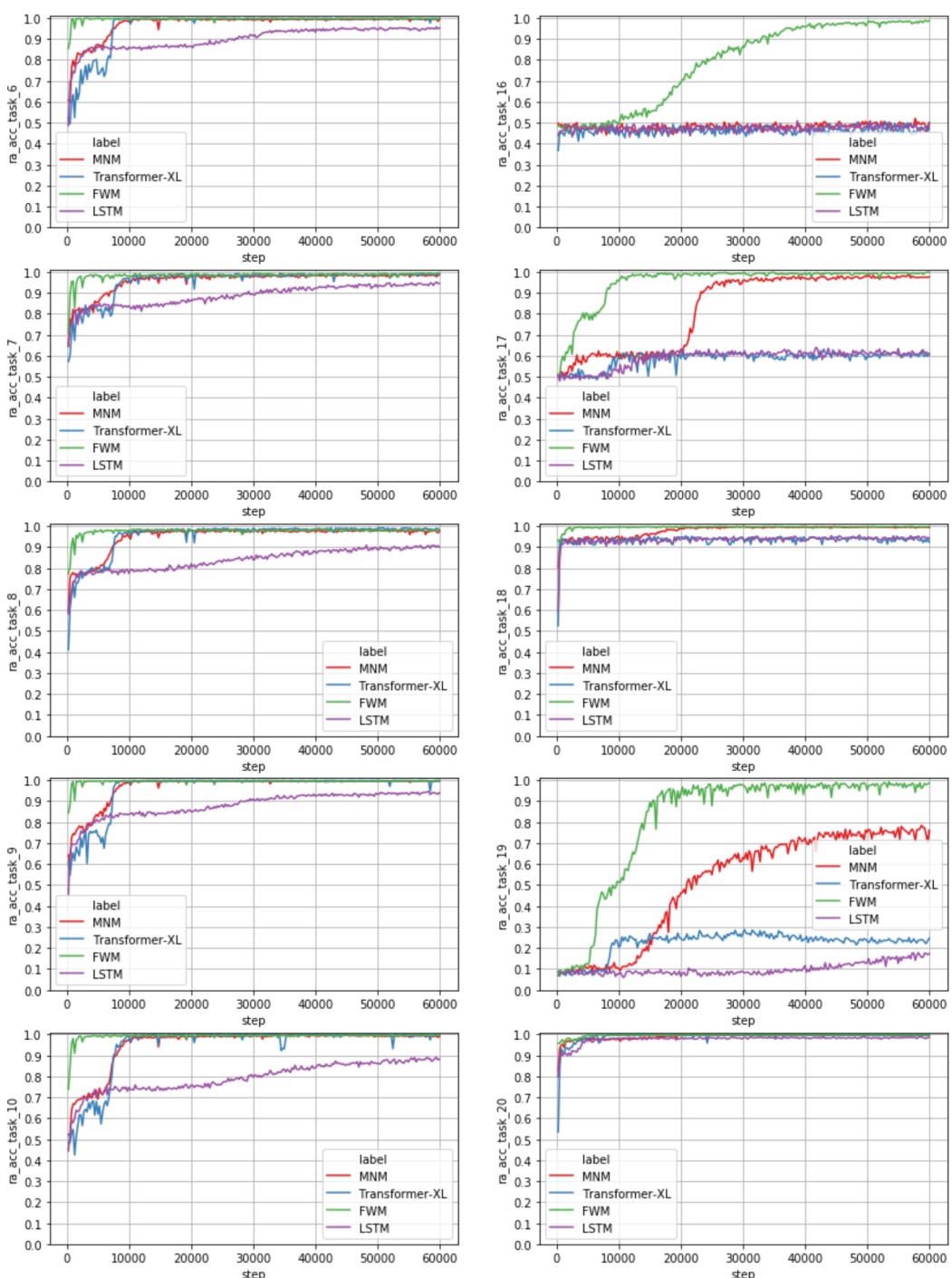

Figure 18: Per-task test set performance comparison of the best catbAbI runs (second part).

## H  Language Modelling

The code of our language modelling experiments is forked from Uber AI Lab's (github.com/uber-research/differentiable-plasticity/tree/master/awd-lstm-lm) which is itself forked from the Salesforce Language model toolkit (github.com/Smerity/awd-lstm-lm). The FWM uses the same three layer LSTM as the slow RNN with the same optimisations as done by Merity et al. (2018). An alternative which we do not explore here is to use multiple FWM-layers each with one LSTM cell and one FWM. We trained our model for 1000 epochs on PTB and 1600 epochs on WT2. Similar to Merity et al. (2018) we switched from Adam to Averaged Stochastic Gradient Descent (ASGD) after 916 epochs and 1372 epochs for PTB and WT2 models respectively. We tune the dropout parameters on the validation set and, after training, we also tune the softmax temperature (tuning the softmax temperature results in ~1 ppl of improvement). The embedding layers were initialized randomly from a uniform distribution, uniform(-0.25, 0.25), which was crucial in our FWM language models. The hyperparameters used for all reported results are in table 4.

The Transformer-XL PTB results were based using the authors official code and hyperparameter setting (see zihangdai.github.io/misc/ptb.zip) which includes AWD-style regularisation, model averaging, and softmax tuning. The WT2 results are based on the same code using the best hyperparameters found by Tim Dettmers (see github.com/TimDettmers/transformer-xl/tree/wikitext2/pytorch).

Table 4: Best hyperparameters of the FWM for our language modelling experiments

| dataset | droupout | dropoute | dropouth | dropouti | wdrop | batch size | ADAM lr | ASGD lr |
|---------|----------|----------|----------|----------|-------|------------|---------|---------|
| PTB     | 0.4      | 0.1      | 0.3      | 0.5      | 0.66  | 20         | 0.001   | 2.0     |
| WT2     | 0.4      | 0.1      | 0.25     | 0.7      | 0.61  | 80         | 0.001   | 0.5     |

### H.1  Results

Table 5: The detailed *evaluation results* of the FWM and Transformer-XL language model for all data partitions of the PTB and WT2 datasets using a batch size of 1. Experiment logs can be found in our git repository.

| model | dataset | seed | loss train | loss valid | loss test | ppl train | ppl valid | ppl test | bits per word train | bits per word valid | bits per word test |
|-------|---------|------|-------|-------|------|-------|-------|-------|--------|--------|--------|
| FWM | PTB | 141 | 2.82 | 4.04 | 4.00 | 16.77 | 56.76 | **54.48** | 4.068 | 5.827 | 5.768 |
|     |     | 142 | 2.66 | 4.05 | 4.01 | 14.26 | 57.43 | 55.17 | 3.834 | 5.844 | 5.786 |
|     |     | 143 | 3.16 | 4.08 | 4.04 | 23.66 | 59.31 | 56.90 | 4.564 | 5.890 | 5.830 |
|     | WT2 | 1881 | 3.32 | 4.23 | 4.18 | 27.80 | 68.74 | 65.07 | 4.797 | 6.103 | 6.024 |
|     |     | 1882 | 2.81 | 4.16 | 4.12 | 16.66 | 63.98 | **61.65** | 4.058 | 6.000 | 5.942 |
|     |     | 1883 | 3.28 | 4.23 | 4.17 | 26.60 | 68.39 | 64.91 | 4.733 | 6.096 | 6.020 |
| TXL | PTB | 2 | 2.87 | 4.09 | 4.04 | 17.62 | 59.71 | 56.63 | 4.139 | 5.900 | 5.824 |
|     |     | 3 | 2.88 | 4.08 | 4.03 | 17.84 | 59.39 | **56.50** | 4.157 | 5.892 | 5.820 |
|     |     | 1111 | 2.86 | 4.09 | 4.03 | 17.52 | 59.73 | 56.53 | 4.131 | 5.900 | 5.821 |
|     | WT2 | 444 | 2.61 | 4.19 | 4.15 | 13.60 | 65.71 | 63.28 | 13.599 | 65.706 | 63.283 |
|     |     | 555 | 2.61 | 4.19 | 4.15 | 13.66 | 65.83 | 63.40 | 13.660 | 65.830 | 63.400 |
|     |     | 666 | 2.61 | 4.14 | 4.19 | 13.62 | 65.73 | **63.11** | 13.622 | 65.725 | 63.109 |

# I  META REINFORCEMENT LEARNING

The meta reinforcement learning experiments trains an agent in training POMDPs and evaluates it on test POMDPs. The environments are directed graphs with labeled edges. As part of the data generating process, novel graphs are sampled according the python algorithm in listing 1. Actions and states are one-hot encoded. The agent receives a 17 dimensional input: the reward location, the current location, the previous action, a fixed bit, the fractional progress as $\frac{\text{current step}}{\text{total steps}}$, and the current reward sum. Getting to the reward location gives a reward of 10. Choosing an invalid action gives a penalty of 0.05. We use a discounting factor of 0.9 and a value coefficient of 0.1. The entropy coefficient of A2C is set to 0.03.

The agent and reward locations are randomly selected at the beginning of the episode. With only 5 states, the reward is reachable in at most 5 steps. As elaborated in section 4.2, such optimal behaviour is only possible once the agent has learned the graphs from its experience. Whenever the reward is placed in the environment a reset timer is set to 0. When the agent reaches the reward, or after 6 unsuccessful steps, the reset timer is set to 0 and the reward and agent are randomly placed in the environment. We train with a batch size of 600 agents and optimize the average step loss using the Adam optimizer.

```python
import numpy as np

def sample_adjacency_matrix(n_actions, n_states, random_state):
  while True:
    A = np.zeros((n_actions, n_states, n_states))

    # every state has to be leavable by at least one action
    for from_state in range(n_states):
      to_state = random_state.choice([i for i in range(n_states)
                                      if i != from_state])
      action = random_state.randint(0, n_actions)
      A[action, from_state, to_state] = 1

    # every state has to be reachable by one or more from-states
    for to_state in range(n_states):
      # only select states which don't have any neighbours given an action
      action_list, from_list = np.where(A.sum(2) == 0)
      # remove self from the selection
      options = np.asarray(list(filter(lambda x: x[0] != to_state,
                                       zip(from_list, action_list))))
      indecies = np.arange(options.shape[0])
      chosen_idx = random_state.choice(indecies)
      from_state, action = options[chosen_idx]
      A[action, from_state, to_state] = 1

    # reject if they are not all connected
    Q = A.sum(0)
    Q[Q > 0] = 1
    for _ in range(n_states):
      Q = np.matmul(Q,Q)
    if (Q == 0).sum() == 0:
      return A
```

Listing 1: Python3 code to sample new environments such that any state is reachable by any other state.

