# OpenReview forum: "Learning Associative Inference Using Fast Weight Memory"
_ICLR.cc/2021/Conference — ICLR 2021 Poster_

### Official Review · AnonReviewer4 · 2020-10-26
**Review of "LEARNING ASSOCIATIVE INFERENCE USING FAST WEIGHT MEMORY"**

**Rating:** 6
**Confidence:** 4

**Review:**

Summary:

The authors present a working memory model composed of a recurrent neural network trained via gradient descent and an associative memory based on the approach taken by Ba et al. (2016) in "Using Fast Weights to Attend to the Recent Past".  The model consists of an LSTM to which takes the input and its own state from the previous step to produce an output (or new state) which is then passed to a fast weight memory (FWM) module.

The application of the fast weights are decomposed into two steps: read and write.  The write step composes the fast weight matrix update where new information is written into F given the LSTM hidden state and the fast weight matrix from the last step.  The read step consists of potentially several  recurrent "inference steps" over the FWM producing an output (e.g. a next step prediction or encoding).

The authors evaluate the model over two separate datasets.  The first is a modified version of the bAbI dataset which concatenates separate bAbI stories together and can be trained and evaluated in either a language modelling (LM) mode or question-answering (QA) mode where knowledge about past facts must be utilized.

Strengths & Weaknesses:

The problem itself is well motivated since associative inference is useful in solving problems that require an accurate working memory.  Fast weight approaches allow us to learn to produce good state representations of the input sequence via slow weights (h_t) and where fast weights provide the associative mechanism to make important links across time.

The authors propose a new model that combines a novel read-write mechanism that relies on a number of inference steps over the fast weights allowing a nice disentanglement of read/write operations taking advantage of the associative inference to both add new relevant associative information (v) while also filtering stale data (v_old).

That said the overall form of the model doesn't seem fundamentally different from what is proposed by Ba et al. (2016) who also used fast weights as a way to attend over past hidden states in combination with a "slow" weighted RNN trained via gradient descent optimization albeit some of the details differ.

Further it would be helpful if the authors could clarify more around the rationale around why particular architectural choices were made.  For instance, why are two keys are generated in the write operation?

Results for both catbAbI don't seem to exceed the performance of the TransformerXL when comparing perplexities in both QA and LM mode and don't exceed TrXL accuracy in LM mode.  However it is noted that the FWM model is in fact much smaller.  It may have been useful to investigate the gated transformer XL which is known to exhibit stronger stability for RL.  Figure 2 is nice though, is there any intuition why the reads vary among strong negative or positive activations as it seems to indicate?

As for the meta-RL problem it would have been nice to see comparisons to baselines other than an LSTM.  For instance, Ritter et al. (2020) in "Rapid Task Solving in Novel Environments" introduce a model that combines an episodic memory with self attention to meta-learn how to explore and exploit navigation to goals in connected graphs.


Other points:

The labelling for the edges Figure 3 isn't really clear.
There's a missing reference in second to last paragraph on page 5: "... QA-mode suffered a TODO% drop in accuracy ..."


Recommendation:

I don't think there's enough here to recommend acceptance.  For starters, I don't think there's quite enough in justification around the architectural choices of the model and exactly what distinguishes this from the model proposed by Ba et al. which also used fast weights in combination with a "slow" weighted RNN.  Next, the results are not strong enough and additional or stronger baselines would have helped paint a better picture of the potential benefits of this approach.  For the results in general, while I think that these results point in the possible direction of the utility of FWM I don't believe the paper in its current form demonstrate that FWM exceeds state of the art in the chosen domains in which it was evaluated.  That said, I believe this is a promising line of research and encourage the authors to try to address the issues raised.

---

> ### Author Response · Authors · 2020-11-13
> **Response to AnonReviewer4**
>
> Thank you for your time and thoughtful critique. In the main response to the reviews, we list a few limitations of the fast weights architecture by Ba et al. (JBFW) as well as some other important changes. Please have a look. We have run experiments on Ba et al. on catbAbI and were unable to find hyperparameters that give any reasonable results. To the best of our knowledge, the JBFW model has never been shown to work well beyond the toy problems that were considered in the original paper. In a workshop paper at NeurIPS 2017 by Schlag et al (see paper references). JBFW are significantly outperformed on a basic associative retrieval task which is similar to the simplest bAbI tasks (task1). We believe those are all good indicators that the differences in the details matter here.
>
> The tensor product in the input pattern of the memory creates a bona fide representational space for the keys learned by the LSTM. This factorisation of the input pattern likely increases the LSTMs representational ability beyond its training distribution. If the LSTM e.g. learns to factorise the key patterns into entities for $k_1$ and locations for $k_2$, then the tensor product $k_1 \otimes k_2$ guarantees a unique vector representation for _any_ key and location pairs which is a requirement for it to be a useful input pattern (minimises interference). To repeat for others: in the extreme, if all keys are orthogonal in their key-space and all locations are orthogonal in their location-space, then all compositions will be orthogonal to each other too. This also includes samples which are technically out-of-distribution and all that is required is that the LSTM can extract the factor representations independently. It is easier to learn such independent functions because there are many samples which share the same factors which can guide the learning of such a function whereas certain pairs are much more rare or even nonexistent. This argument has been introduced by Schlag et. al. (2018) so we only briefly mention it.
>
> As we have mentioned in our general response: we had a bug in our code and fixing this resulted in a performance increase across all models. The FWM is now convincingly beating all other models with a lead of 7.9% averaged over 3 seeds. In comparison with the TXL, our FWM is not only smaller but also requires many fewer activations to be kept in memory (see table 1).
>
> We are not familiar with the Transformer architecture applied to RL. Do you refer to "Stabilizing Transformers for Reinforcement Learning" by Parisotto et al. (ICML 2019)? There does not seem to an official public code repository for this work.
>
> We think that due to the simplicity of the linear hetero-associative memory in this network, the sign of the dot product between the write patterns and the read keys doesn't matter as long as it is always the same for a certain pattern. What is more important is the absolute value of the dot product.
>
> Thank you for mentioning the work of Ritter et al. (2020). This looks interesting indeed but is also missing an official implementation. It appears to us that it would require a disproportional amount of work to incorporate it as a baseline. Our RL experiments are mostly for demonstrating the versatility of our method. We hope that future work in RL will evaluate different memory augmented models in more complex environments.
>
> Thanks for pointing out the typo! In the latest version of the submission, we have made significant improvements to the text and the figures.
>
> With the latest improvements in our work, we believe that we have addressed the main issues raised in your review and we encourage you to reevaluate your decision. Please let us know if you have any further questions or comments.

---

> > ### Comment · AnonReviewer4 · 2020-11-24
> > **New Experimental results strengthen the claims made**
> >
> > Thanks for your thoughtful rebuttal, I think this clarifies a number of things for me and the new experimental result strengthen the argument.  I still believe an ablation or meta-learning baseline would be a useful comparison for the meta-learning experiments. However, there are many improvements in the latest draft which address my greatest concerns.  These include your points about the baselines, clearer figures, clarification of improvements over earlier fast weight models, the clarity around the architecture and including the location and content based keys and, last but not least, the bug fix which now clearly demonstrates an improvement over the chosen baselines, I'm happy to increase my score for this paper to a six.

---

> > > ### Author Response · Authors · 2020-11-24
> > > **Thanks**
> > >
> > > Thank you, AnonReviewer4, for taking the time to reassess our submission.

---

### Official Review · AnonReviewer1 · 2020-10-28
**Interesting method for fast memory, but the experiments are not totally convincing.**

**Rating:** 7
**Confidence:** 4

**Review:**

This paper presents a new method called Fast Weights Memory (FWM) to add an associative memory to an LSTM.

Model:
* FWM updates its fast weights through a differentiable perceptron like update at every step of an input sequence. The slow weights of the LSTM are instead updated only during training using gradient descent.
* FWM is based on previous work: TPR (Tensor Product Representation). TPR is a mechanism that uses tensor products to generate unique representations of combination of components.
* For long sequences FWM also has specialized components that allow it to update deprecated associations.

FWM is related to:
* TPR-RNN: a sentence-level model for reasoning on text, achieving excellent results on bAbI.
* MNM (Metalearned Neural Memory): a word-level model which augments an LSTM with a FFNN as its memory, trained with a meta-learning objective.

The authors propose the new task "catbAbI", a variation of the existing task "bAbI".  catbAbI seems to be mostly just a concatenation of the stories, questions and answers in bAbI into a single textual sequence. It's unclear how much harder catbAbI is compare to bAbI in principle.

TPR-RNN and MNM are only trained for short sequences and so will have a hard time on catbAbI. The authors show that MNM in particular does poorly on the long sequences in catbAbI.

Results:
* good performance on catbAbI (language reasoning) -- but this is a new task, so no real baselines in other papers.
* good results meta-reinforcement-learning for POMDPs compared to LSTMs.
* good results on PTB language models, better than other published models, but not state of the art.

Limitations:
* FWM requires an order 3 tensor, which scales poorly in both time and space computational complexity. This limits this work to relatively small models.

Questions:
* catbAbi simply converts bAbI into a single sequence of tokens. Does this really increase the true difficulty of the task, or is it rather a way of artificially limiting the class of models used to solve the task to simple LM-like models? Is it possible to reconstruct bAbI from catbAbI with simple heuristics?
* Could you report results for FWM on bAbI? It’s pretty unclear at the moment how to compare the results on bAbI of FMW to the ones e.g. in cited “Metalearned Neural Memory” paper. Or at least results on a version of bAbI where predictions are run for each story separately, so that MNM is not as penalized for not being able to deal with long sequences of text.
* In figure 2, what does the color represent?

Typos:
* Page 2:
   * *A biologically more plausible
   * *stateful weights that can adapt
   * Most memory-augmented NNs are based *on content-based…
* Page 3:
   * becomes a part of the *model's output.
   * Figure 1: A simplified illustration of *our proposed method
   * third-order tensor operations using *matrix multiplications
* Page 4:
   * Wq → Wn in equation (1)
* Page 5:
   * there is TODO left

---

> ### Author Response · Authors · 2020-11-13
> **Response to AnonReviewer1**
>
> Thank you for your detailed review. Please have a look at the general response to the reviews as there have been some important changes that are relevant to your review.
>
> We added two ablation experiments to the appendix which demonstrates how the FWM drops in accuracy if the key vectors are concatenated. While it is a notable drop in performance, it still outperforms the Transformer-XL. It is true that the presented FWM scales cubic but it is independent of its sequence length! The Transformer-XL, or any Transformer, stores all keys and values of previous steps that are within its context window which often comes with a large memory requirement. Compare e.g. the number of activations in table 1 between the TXL and the FWM!
>
> Q1: It does, because the model needs to make sure it is not mixing facts from previous stories with the facts of the ongoing story. This means that the model needs to learn to update its memory accordingly. Regular bAbI, on the other hand, is often simplified to "sequence classification". In the appendix section A, we discuss this at length. Please have a look if you have not seen it yet.
>
> Q2: We directly compare with MNM in our work. Our codebase is focused on catbAbI but we'll check if we can add those results in the near future.
>
> Q3: We have updated the caption of that figure to hopefully better explain the visualisation. The colour represents the dot product of the write keys of previous steps $k_1 \otimes k_2$ and the query $n \otimes e$ at the "?" position at which the answer has to be predicted.
>
> Thank you very much for pointing out typos! We have fixed those (and several others) throughout our text. We hope that the latest changes have increased your confidence in our work.

---

### Official Review · AnonReviewer3 · 2020-10-28

**Rating:** 7
**Confidence:** 4

**Review:**

The solution proposed is the combination of an RNN (LSTM) and Fast Weighted Memory (FWM). The LSTM produces a query to the memory used to retrieve information from the memory and be presented at the model output. It also controls the memory through fast weights that are updated through a Hebbian mechanism. The FWM is based on Tensor Product Representations (TPR). The FWM is differentiable and builds upon the work of TPR-RNN from Schlag and Schmidhuber and Metalearned Neural Memory (MNM) by Munkhdalai et al. In the experimental section, the authors propose a concatenated version of the bAbI dataset to test their model with language modeling and question answering. Further the model is trained on a meta-learning task over POMDPs on graphs, and on language modeling on the PennTree Bank dataset. They show that the LSTM-FWM model generalizes better than without memory and similar models and with smaller capacity.

======================================

Indeed, the FWM model is relevant to this community and involves current scientific discussion and challenges. The paper is clear and is enjoyable to read. Math derivations and experimental results seem sound. Nevertheless, there are some clarity issues with the PTB language modeling task.

======================================

Would appreciate if the authors can answer to the following questions:

How is the FWM (tensor $\mathbf{F}_t$) initialized? How does the initialization influence training and performance?

How is Nr selected?

What is the vocabulary size in catbAbI? Is the embedding layer learned or pre-trained?

“The experimental results in table 2 demonstrate a relative improvement over the AWD- LSTM baselines, which suggest the benefit of our FWM.” It is unclear what is the benefit in the PTB dataset. The results show that the LSTM model has slightly better perplexity (60.0 / 57.3) than the LSTM-FWM (61.39 / 59.37). Please, could you clarify the above note versus the numbers?

Does Figure 2 have missing details? The caption doesn’t seem to match the figure or it is unclear what authors are referring to.

Figure 3 can benefit from using a bigger font for the node and edge values.

======================================

I'm inclined to accepting this paper. I found the idea simple but yet effective, and tested correctly in the experimental sections. Would appreciate it if the authors can improve the clarity surrounding Figure 2, and explain the misleading comment regarding the PTB task.

======================================

Minor issues:

-Page 2: “An biologically” -> “A biologically”

-Page 2: “pattern is is different” -> “pattern is different”

-Page 5: Please correct with the missing number “suffered a TODO% drop”

-Page 5: “figure 4.1.1” -> “Figure 2”

-Page 6: “noteable” -> “notable”

==================================
UPDATE

Thank you for replying to my questions and clarifying in the document.

---

> ### Author Response · Authors · 2020-11-13
> **Response to AnonReviewer3**
>
> Thank you for your review. Your summary is correct but please have a look at our recent changes in our general response to the reviews. Here we'll directly respond to your individual questions and comments.
>
> Q1: The FWM is initialised with zeros. We did not experiment with training the initial fast weights.
>
> Q2: $N_r$ is a hyperparameter and was selected based on the number of inference steps that is probably necessary to solve all tasks in bAbI. We have added an ablation experiment to the appendix (section E figure 7) which shows a drop in performance with fewer read operations.
>
> Q3: The vocab size of catbAbI is 200 and the word-embeddings are learned.
>
> Q4: This was a typo. We have now PTB and WT2 experiments with the appropriate hyperparameter tuning and we now beat both baselines. We also added figure 7 which gives an example of how the FWM improves over the AWD-LSTM on PTB.
>
> Q5: We have rewritten the caption of figure 2 to be easier to understand. We hope this new version clears up any remaining confusion.
>
> We have fixed the typos you found (and several others) and improved the quality of various figures (including figure 3). Thank you for your positive feedback.

---

### Official Review · AnonReviewer2 · 2020-11-02
**The work proposes to complement an LSTM with an additional associative memory model with fast changing weights. The proposed combination demonstrates good results on several ML tasks.**

**Rating:** 7
**Confidence:** 3

**Review:**

This looks like an interesting paper with an original proposal. The empirical results on synthetic tasks are also good. The main problem that I am having is with the proposed network, specifically equation 3. I do not see why it makes sense to consider an outer product of $n$ and $e$ as an argument to FWM. As is mentioned in the paper (in appendix A) it would make more sense (both conceptually and from the perspective of complexity) to concatenate those two inputs, or even consider two separate inputs for the associative memory module. The authors argue that in that case the memories would interfere with each other. This is true if a weak associative memory, like the one considered in this work is used. However, if the authors used a modern Hopfield network such an interference would not be a problem. Specifically, consider the situation when after applying the FWM weights to $n$ and $e$ the results are passed through a steep non-linear activation function, like in Ref [1] (see for instance formula 10). This would suppress the interference between the memories and provide a nice memory recovery. Additionally, with these “stronger” models of associative memory the key vectors do not have to be orthogonal.

Experimental results look fine, however, I think the work would benefit from some comparisons with other proposals for fast changing weights models, for example Ref [2].

I am not sure I understand the last paragraph on page 2. It is very easy to convert Modern Hopfield Networks from the autoassociative to heteroassociative type. One just needs to introduce additional matrices for queries, keys and values, like it is done in Ref [3] when comparing Modern Hopfield Networks with attention. Also, when referring to Modern Hopfield Networks, the reference for the original work, Ref [1], is missing.

A couple of presentational suggestions:

1. Figure 1 seem to be inaccurate. In order to generate x_{t+1} one needs to take into account both the output of FWM and current state h_t. Only the first arrow is shown in figure 1.

2. After equation 4, what is W_n? Looks like a misprint - should it be W_q? Also in the second line after equation 4 there are some misprints in the formulas.

I also have some questions:

1. Typically most associative memory models converge to a fixed point if one runs them for a long time. It is not obvious to me if dynamical rules described by equations 1-3 converge to a fixed point after a sufficiently large number Nr of iterations. Do they converge to a fixed point or not?

2. It looks to me that the results reported in table 2 indicate that LSTM without FWM have lower perplexity than LSTM with FWM on that task. At the same time, the authors seem to say in the text (second paragraph on page 8) the opposite. Could the authors please clarify this?

I am willing to increase the scores for this submission if the questions/comments above are addressed.

References:

[1] Krotov and Hopfield, NeurIPS 2016. Dense associative memory for pattern recognition, arXiv:1606.01164.

[2] Ba, et al, NeurIPS 2016, Using fast weights to attend to the recent past, arXiv:1610.06258.

[3] Ramsauer, et al., 2020. Hopfield networks is all you need, arXiv:2008.02217.

---

> ### Author Response · Authors · 2020-11-13
> **Response to AnonReviewer2**
>
> Thank you AnonReviewer2 for your comments. We have created a general response to the reviews, please have a look if you have not seen it yet. Here we'll address your more individual comments. We have also added an ablation experiment to the appendix and improved the text of the subsections about equation 3.
>
> The tensor product in the input pattern of the memory creates a bona fide representational space for the keys learned by the LSTM. This factorisation of the input pattern likely increases the LSTMs representational ability beyond its training distribution. If the LSTM e.g. learns to factorise the key patterns into entities for $k_1$ and locations for $k_2$, then the tensor product $k_1 \otimes k_2$ guarantees a unique vector representation for _any_ key and location pairs which is a requirement for it to be a useful input pattern (minimises interference). To repeat for others: in the extreme, if all keys are orthogonal in their key-space and all locations are orthogonal in their location-space, then all compositions will be orthogonal to each other too. This also includes samples which are technically out-of-distribution and all that is required is that the LSTM can extract the factor representations independently. It is easier to learn such independent functions because there are many samples which share the same factors which can guide the learning of such a function whereas certain pairs are much more rare or even nonexistent. This argument has been introduced by Schlag et. al. (2018) so we only briefly mention it.
>
> We acknowledge in our work that the associative memory used is a rather simple one. Though we'd like to point out that it has to be constructed, edited, and read with representations generated by an LSTM. This is an added layer of complexity. Other work on associative networks is usually about storing a fixed set of patterns. In our opinion, extending it to modern Hopfield networks is not as trivial as adding a steep non-linear activation function. This is because we superimpose previous keys and values in the fast weight matrix. The Eq. 10 in Krotov and Hopfield (2016) requires access to every previous keys and values to compute the mixing coefficient. Replacing the query with the keys weighted by that mixing coefficient is the iterative mechanism which allows it to converge to one of the keys (the fixpoints). Using a non-linearity like the softmax then results in an attention mechanism which is in its essence equivalent to the Transformer attention but which would also grow with the length of the sequence (or the number of samples to store). The FWM instead accumulates all previous updates into its Fast Weight tensor.
>
> It is correct that the auto-associative type can be converted into a hetero-associative type if it's input and output domain is subdivided and properly trained/constructed. However, our memory is controlled by the LSTM and no such constraints are explicitly applied. We believe that the explicit separation of the domain and codomain is a useful bias for the problems considered in this work.
>
> We have added the missing reference, fixed the typos, and edited the figure description that you mention. Thank you for those details!
>
> Q1: The $N_r$ steps are not thought of minimising the energy landscape of the memory but are instead $N_r$ independent queries. This is e.g. demonstrated in figure 3 where we can see how the query-pattern matches the write-pattern at different previous steps. The query is a single step and no convergence analysis can be given because we do not have access to each key separately at time step t.
>
> Q2: Yes, this was a mistake in the text. However, we added new results based on our hyperparameter tuning which has the FWM now beat both baselines.
>
> We hope that we have addressed your questions and comments adequately. Please let us know if you have any further questions or comments.

---

> > ### Comment · AnonReviewer2 · 2020-11-24
> > **Post revision comments**
> >
> > Thank you for answering my questions. I have read the revised paper, and I think it looks better now. It’s nice to see that the numbers in Table 2 improved after tuning the hyperparameters. I have updated my score.

---

### Author Response · Authors · 2020-11-13
**Rebuttal: General Response**

We'd like to thank the reviewers for their helpful feedback! Since our initial submission, there have been a few big and several small changes. We think that the newest version of the manuscript is a massive improvement in outcome and overall quality and we hope the reviewers will find the time to appreciate the change.

These are the major differences we'd like reviewers to be aware of:

1.) We have found a bug in our catbAbI code related to how states are carried between epochs. After fixing this issue, we reran the hyperparameter search for all our models. We see improvements in all models with **our FWM now at an average of 96.75% test accuracy** (see table 1)! The learning curves of the best seed for each model further visualises the gap between the FWM and our other models (figure 2).

2.) We added WT2 to the language modelling results and our own Transformer-XL baseline. After tuning the hyperparameters **we now beat the AWD-LSTM and the AWD-Transformer-XL on PTB and WT2** (see table 2).

3.) Reviewer 1 and 4 have both scrutinised the connection with the Fast Weight RNN by Ba et al. (JBFW). Notice, that we did mention that we ran JBFW models but decided to exclude it since we were not able to find any hyperparameters that converge. To the best of our knowledge, the JBFW model has never shown to work on any shared benchmark (like e.g. bAbI or language modelling). We don't find this surprising as it has some obvious technical issues like e.g.:
- It is essentially just a classic Elman RNN and likely suffers from vanishing gradients when applied to long sequences.
- It only adds to its fast weights. It is difficult to understand how previous information is removed or updated.
- It has a fixed fast weight decay mechanism, making it impossible (by design) to store information for many steps.
- Its memory is updated with the outer product $h_t \otimes h_t$ which, as in a Hopfield network, allows to retrieve $h_t$ from a noisy version of itself ~$h_t$. It is, in theory, possible to convert Hopfield networks to the hetero-associative type, but we believe explicitly constructing hetero-associative memories is in practice much easier to learn.

4.) We rearranged and improved the subsection on the writing and reading mechanism which now more intuitively explains our update rule. We also added a proof to the appendix which derives the update rule and rewrote the description of figure 3 regarding the visualisation of how the FWM chains independent facts to be easier to understand.

5.) We have added two ablation experiments. One w.r.t. to $N_r$ (the number of read operations) which results in a performance decrease (see figure 7 in appendix E) and a concatenation of the key vectors which also results in a performance drop (figure 8 in appendix E). We now refer to those ablations in the discussion section.

6.) We have improved the quality of the document by fixing typos, editing various sentences, and improving the presentation of most figures.

Further details can be found in the responses to each reviewer.

---

### Decision · Program_Chairs · 2021-01-07
**Final Decision**

**Decision:**

Accept (Poster)

**Comment:**

This paper proposed a way to combine LSTMs with Fast weights for associative inference.

While reviewers had concerns about comparison with Ba et al., and experimental results, the authors addressed all the concerns and convinced the reviewers. The revision strengthened the paper significantly. I recommend an accept.